# NEK7 regulates dendrite morphogenesis in neurons via Eg5-dependent microtubule stabilization

Francisco Freixo[1], Paula Martinez Delgado[2], Yasmina Manso[3,4,5,6], Carlos Sánchez-Huertas[1,7], Cristina Lacasa[1], Eduardo Soriano[3,4,5,6], Joan Roig [2] & Jens Lüders [1]

Organization of microtubules into ordered arrays is best understood in mitotic systems, but remains poorly characterized in postmitotic cells such as neurons. By analyzing the cycling cell microtubule cytoskeleton proteome through expression profiling and targeted RNAi screening for candidates with roles in neurons, we have identified the mitotic kinase NEK7. We show that NEK7 regulates dendrite morphogenesis in vitro and in vivo. NEK7 kinase activity is required for dendrite growth and branching, as well as spine formation and morphology. NEK7 regulates these processes in part through phosphorylation of the kinesin Eg5/KIF11, promoting its accumulation on microtubules in distal dendrites. Here, Eg5 limits retrograde microtubule polymerization, which is inhibitory to dendrite growth and branching. Eg5 exerts this effect through microtubule stabilization, independent of its motor activity. This work establishes NEK7 as a general regulator of the microtubule cytoskeleton, controlling essential processes in both mitotic cells and postmitotic neurons.

[1] Institute for Research in Biomedicine (IRB Barcelona), The Barcelona Institute of Science and Technology, Baldiri Reixac 10, 08028 Barcelona, Spain. [2] Molecular Biology Institute of Barcelona (IBMB-CSIC), 08028 Barcelona, Spain. [3] Department of Cell Biology, Physiology and Immunology, Faculty of Biology, Institute of Neurosciences, University of Barcelona, 08028 Barcelona, Spain. [4] Centro de Investigacion Biomedica en Red sobre Enfermedades Neurodegenerativas (CIBERNED), ISCIII, 28031 Madrid, Spain. [5] Vall d'Hebron Institute of Research, 08035 Barcelona, Spain. [6] Institucio Catalana de Recerca i Estudis Avancats (ICREA), 08010 Barcelona, Spain. [7] Present address: Centre de Recherche en Biologie Cellulaire de Montpellier, CRBM-CNRS, UMR5237 Montpellier, 1919 Route de Mende, 34293 Montpellier, France. Correspondence and requests for materials should be addressed to J.Lüd. (email: jens.luders@irbbarcelona.org)

The compartmentalization of neurons into morphologically and functionally distinct domains, including cell body, one thin and long axon, and multiple shorter and tapered dendrites, is essential for neuronal function and connectivity. Excitatory synaptic input from other neurons is received through the somato-dendritic compartment via protrusions termed spines, is integrated in the cell body and then propagated through the axon to the presynaptic terminal, where the axon makes contact with other neurons.

The identities of the axonal and dendritic compartments are largely determined by the differential organization of microtubule arrays, involving distinct sets of motors and other microtubule-associated proteins, to allow transport of specific cargo and with specific directionality[1–3]. Whereas microtubules in axons are arranged in parallel bundles with almost uniform plus-end-out orientation, dendrites contain bundled microtubules with mixed polarity or predominant minus-end-out orientation, depending on the organism[3]. These differential microtubule configurations are established early during the formation of axon and dendrites[4]. However, despite these differences, dendrite growth and branching at the distal tips requires, as in axons, supply of predominantly anterograde (plus-end-out) microtubules[5]. Subsequent to their incorporation at the tips of the growing processes, microtubules become stabilized and less dynamic. Whereas in axons essentially all stable microtubules are of plus-end-out orientation, microtubule stabilization in dendrites occurs mostly in the minus-end-out microtubule population[4]. While the basic configuration of microtubules within neuronal compartments has been fairly well described, the molecular mechanisms by which these arrays are established, selectively stabilized, and maintained remain poorly understood. Indeed, most of our insight into the molecular basis of microtubule organization stems from mitotic systems.

Mitotic spindle assembly is a process of immense complexity, but extensive functional screening has identified many if not most of the important players[6–9]. Moreover, the proteomes of key structures of the microtubule cytoskeleton in cycling cells such as the centrosome or the mitotic spindle have been identified[10–14]. Extrapolating this insight into understanding microtubule organization in postmitotic cell types such as neurons is not straightforward though, since the microtubule organizing structures and the gene expression profiles in postmitotic cells differ from those in cycling cells. During neuron differentiation, for example, the centrosome gradually loses its role as microtubule organizer[15,16] and for the majority of mitotic microtubule regulators, it is not known whether they are also present and contribute to microtubule organization in neurons. However, in recent years several so-called mitotic factors were shown to have such dual function. Examples are various motor proteins[17,18], the nucleator γTuRC[15,16,19,20], the microtubule branching factor augmin[15], the spindle assembly factor TPX2[21–23], and the kinase Aurora A[23].

Here, we analyzed candidate microtubule regulators in cultured neurons by expression profiling and targeted RNA interference (RNAi) screening and identified the kinase NEK7 as novel regulator of dendrite morphogenesis in vitro and in vivo. NEK7 controls dendrite growth and branching as well as the formation and morphology of spines. NEK7 has been described previously as a mitotic kinase with roles in spindle assembly and mitotic progression[24,25]. It drives separation of the duplicated centrosomes at mitotic prophase, by promoting accumulation of the kinesin motor Eg5/KIF11 around centrosomes[26], and promotes cytokinesis by regulating localization of the kinesin KIF14 to the spindle midzone in late mitosis[27]. Interestingly, we found that the role of NEK7 in neurons is at least in part also mediated by Eg5. Thus, the NEK7–Eg5 module exemplifies the recycling of mitotic microtubule regulators to carry out similar functions in postmitotic cells.

## Results

**Identification of microtubule regulators in neurons.** To identify genes that may contribute to microtubule organization in neurons we took advantage of proteomic data previously obtained from cycling cells. Using information from the literature and from public databases we compiled a list of "microtubule cytoskeleton-associated" (MCA) genes (Supplementary Data 1). These genes encode proteins that in cycling cells were shown to associate with structures of the microtubule cytoskeleton including centrosomes, kinetochores, mitotic spindles, and midbodies, and/or have roles in mitotic spindle assembly and function (http://microkit.biocuckoo.org; http://centrosome.cnb.csic.es/human/centrosome)[6,7,9,10,12,14,28–30]. We then performed whole genome microarray analysis (20,398 genes including 1456 MCA genes) of cultured mouse hippocampal neurons at different stages of differentiation and maturation (0, 1, 3, 6, 12, and 15 days in vitro (DIV)) (Fig. 1a; Supplementary Data 1). Cluster analysis of differentially expressed genes revealed groups of genes with similar expression profiles during the time course of neuronal culture (Fig. 1a; Supplementary Data 2). Of a total of 377 differentially expressed MCA genes, 173 (~46%) were overall downregulated (Fig. 1a; Supplementary Data 2; clusters 1, 4, 6, 7, 8), 182 (~48%) were upregulated (Fig. 1a; Supplementary Data 2; clusters 2, 3, 9, 11, 12, 14), and 22 (~6%) showed transient changes in expression (Fig.1a; Supplementary Data 2; clusters 5, 10, 15, 16, 17, 18, 19). We then identified various marker genes and confirmed that they displayed the expected expression profiles. For example, we observed downregulation of genes with known roles in proliferating cells such as DNA topoisomerase *Top2b* or the chromatin assembly factor *Nasp* (Fig. 1a, cluster 4; Supplementary Data 2), and upregulation of genes with important functions in postmitotic neurons such as *Snap25*, encoding a SNARE protein regulating neurotransmitter release, or the brain-specific cell adhesion molecule *Ncam2* (Fig. 1a, cluster 2; Supplementary Data 2), similar to what was observed in a previous study[31]. Among the differentially expressed MCA genes, only upregulated genes were considered as candidates for further analysis.

To evaluate the validity of the microarray results we performed quantitative real-time PCR (qRT-PCR) analysis on a subset of upregulated candidates and, as controls, also on some genes that did not display differential expression. Candidates were shortlisted based on evidence from previous studies supporting their physical and/or functional association with the microtubule cytoskeleton. Further, we only included candidates that had not been suggested previously to have a role in neurons. Indeed, when we compared expression changes between 0 DIV and 15 DIV, the qRT-PCR result was overall very consistent with the microarray data (Supplementary Fig. 1a, b).

We then performed RNAi screening on selected candidates in cultured mouse hippocampal neurons, scoring for any morphological defects during neuron differentiation, which led to the identification of *Nek7*. Western blotting confirmed that NEK7 protein was present in cultured neurons from 0 to 21 DIV, displaying upregulation at late stages (Fig. 1b, c) and in extracts of mouse cortex throughout development (e18.5 to P30) with increased expression at P15/P30 compared to earlier stages (Fig. 1d, e). These results are in agreement with our microarray and qRT-PCR data and with previous in situ hybridization results that detected *Nek7* expression in the brain[32].

**Knockdown of NEK7 reveals roles in dendrite morphogenesis.** For the functional analysis in neurons we identified small hairpin RNAs (shRNAs) that allowed efficient depletion of NEK7 as determined by western blotting (Fig. 1g). We then transfected shRNA plasmids together with a plasmid expressing enhanced

green fluorescent protein (EGFP) into hippocampal neurons either at 1 DIV or at 7 DIV and fixed at 5 DIV or 14 DIV, respectively. The integrity of the neuronal microtubule cytoskeleton is crucial for axonal and dendritic growth[15,16,19]. Thus, as readout for potential alterations in the microtubule cytoskeleton, we traced the morphology of GFP-positive neurons in combination with microtubule-associated protein-2 (MAP2) staining and quantified total lengths of

axon (GFP-positive/MAP2-negative) and dendrites (GFP-positive/MAP2-positive). In NEK7-depleted neurons at 5 DIV, there was no significant difference in total dendrite length but a slight increase in axon length compared to controls (Fig. 1i; Supplementary Fig. 1c, e). The strongest phenotype, however, was observed at later stages, when NEK7 levels are increased. Surprisingly, at 14 DIV, NEK7 depletion did not affect axon length as in earlier stages, but resulted

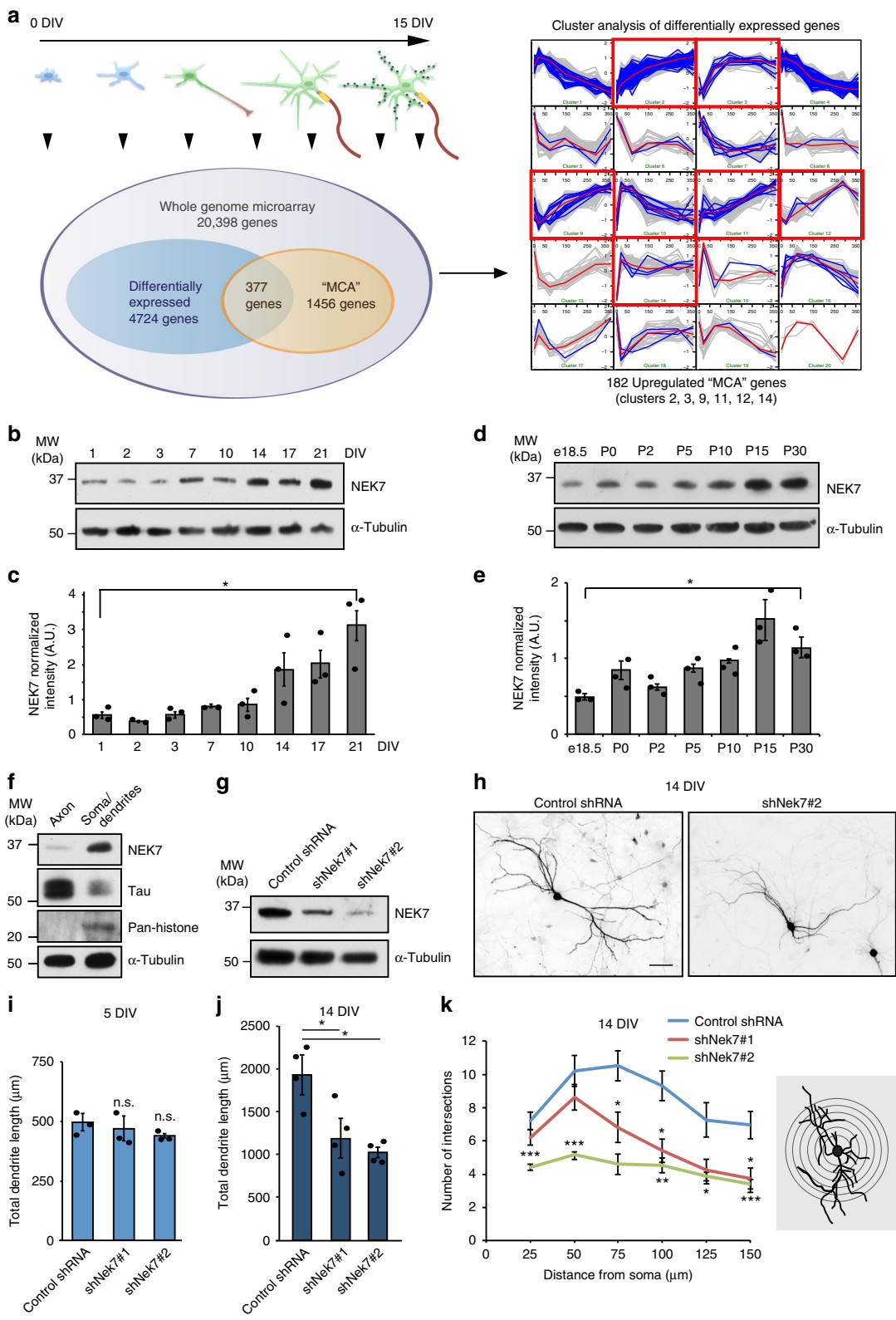

in an ~40–50% reduction in total dendrite length, suggesting that NEK7 has a specific and important role in dendrites of more mature neurons (Fig. 1h, j; Supplementary Fig. 1d). Indeed, dendritic arborization was also strongly reduced in NEK7-depleted neurons compared to controls (Fig. 1h, k). In agreement with this finding, probing of axonal and somato-dendritic cellular fractions from neurons at 9 DIV with NEK7 antibodies revealed an enrichment of NEK7 in the somato-dendritic compartment (Fig. 1f). To further corroborate that NEK7 did not affect dendrite length and arborization at earlier stages, we trypsinized and replated neurons previously transfected with *Nek7* shRNA from 2 to 5 DIV, which allowed the study of early neurite outgrowth under conditions of extended NEK7 depletion. Confirming our previous experiment, 2 days after replating we observed a slight increase in total axon length compared to controls, whereas there was no effect on dendrite length (Supplementary Fig. 1f, g). Altogether, our results suggest that a major role of NEK7 in neurons is the regulation of dendrite growth and arborization at later stages of differentiation.

Since in cycling cells NEK7 forms a signaling module with the highly similar NEK6 and the activating kinase NEK9[33], we asked whether these two kinases participate in the regulation of dendrite morphogenesis. While NEK9 expression in hippocampal neurons was relatively constant throughout differentiation, NEK6 expression was increased between 7 and 10 DIV (Supplementary Fig. 2a). Importantly, levels of NEK6 or NEK9 were not affected by depletion of NEK7, and NEK7 levels were not affected by depletion of NEK6 or NEK9 (Supplementary Fig. 2b). Neurons were then transfected with these knockdown constructs at 7 DIV, and analyzed for total dendrite length at 14 DIV. Neither depletion of NEK6 nor depletion of NEK9 had any significant effect on dendrite length (Supplementary Fig. 2c), suggesting that NEK7 controls dendrite length independently of these kinases.

**Dendrite morphogenesis requires NEK7 kinase activity**. To address whether the regulation of dendrite length required NEK7 kinase activity, we generated shRNA-resistant versions of FLAG-tagged wild-type NEK7 and catalytically inactive NEK7 D179A[34] (Supplementary Fig. 2f, g). Neurons were transfected at 7 DIV and analyzed for dendrite length and branching at 14 DIV. As expected, FLAG-NEK7 but not FLAG-GFP was able to rescue the reduced dendrite length caused by NEK7 depletion (Fig. 2a, b). However, the decrease in dendrite length was not rescued by the FLAG-NEK7 D179A mutant, indicating that rescue required NEK7 kinase activity (Fig. 2a, b). Similarly, only wild-type FLAG-NEK7 but not the FLAG-NEK7 D179A mutant rescued the reduced complexity of the dendritic arbors in NEK7-depleted neurons (Fig. 2a, c). Interestingly, whereas overexpression of wild-type FLAG-NEK7 or a constitutively active NEK7 Y97A

mutant (mutant in the autoinhibitory residue Y97)[35] did not alter dendrite length, expression of the inactive FLAG-NEK7 D179A mutant slightly decreased dendrite length (~2200 µm compared to ~2600 µm in control cells; Supplementary Fig. 2h), consistent with a dominant-negative effect. Indeed, a similar mutant (NEK7 D179N) was previously reported to exert a dominant-negative effect in cycling cells, in that context potentially by sequestering NEK9, the kinase that activates NEK7 in mitosis[34].

Considering the importance of the dendritic compartment in receiving synaptic input, we also analyzed whether NEK7 depletion affected the number and morphology of spines, small post-synaptic protrusions that receive excitatory input. Imaging of neurons at 14 DIV showed that NEK7 depletion caused a decrease in spine density by ~50% in the principal dendrite compared to controls, which was fully rescued by co-expression of FLAG-NEK7 but only partially by the kinase-dead mutant NEK7 D179A (Fig. 2d, e). NEK7 depletion also affected spine morphology: mushroom-shaped spines were rare and most had filopodia-like appearance (spine head diameter 0.32 µm compared to 0.43 µm in controls) (Fig. 2d, f). The spine defects were rescued by co-expression of FLAG-NEK7 but not FLAG-NEK7 D179A (Fig. 2d–f). We also analyzed spines in neurons depleted of NEK6 or NEK9, but could not detect any defects (Supplementary Fig. 2d, e). Together, the results indicate that NEK7 kinase activity is required for morphological differentiation of dendrites at multiple levels including formation and morphogenesis of spines.

**Eg5 inhibition phenocopies NEK7 depletion in mature neurons**. A known substrate of NEK7 in cycling cells is the kinesin Eg5/KIF11[26]. Since several previous studies have also implicated Eg5 in axon and dendrite growth in neurons[17,22,36], we speculated that NEK7 might control dendrite and spine morphogenesis through Eg5. Consistently, Eg5 expression peaked between 7 and 10 DIV, a period that corresponds to dendritic maturation[37], and Eg5 was enriched in the somato-dendritic compartment of 9 DIV neurons (Supplementary Fig. 3a, h).

If Eg5 was the effector downstream of NEK7 we expected Eg5 inhibition to phenocopy the effects of NEK7 depletion. To test this we treated EGFP-transfected neurons with the Eg5 inhibitors monastrol or S-trityl-L-cysteine (STLC) at 7 DIV and then fixed and immunostained cells at 14 DIV. Using EGFP as morphological marker, we first measured total dendrite length and branching. In agreement with previous reports[36], treatment with monastrol or STLC significantly decreased total dendrite length (1067 µm and 1208 µm, respectively, compared to 1757 µm in control neurons) (Supplementary Fig. 3b, c). This effect was specific to Eg5 inhibition since it was also observed in cells depleted of Eg5 by shRNA (Supplementary Fig. 3i, j). Both inhibitors also reduced dendrite

**Fig. 1** Identification of NEK7 as regulator of dendrite length and branching. **a** (Left) Cultured neurons at the depicted stages were subjected to microarray analysis (black arrowheads indicate sampling time points). Of the 1456 microtubule cytoskeleton-associated (MCA) genes, 377 were differentially expressed. (Right) Cluster analysis of differentially expressed genes. Expression profiles of non-MCA and MCA genes are depicted in gray and blue, respectively. Mean expression profiles are shown in red. Red boxes highlight overall upregulated clusters. **b** Hippocampal culture lysates from 1 to 21 DIV were immunoblotted against NEK7, and α-tubulin as loading control. **c** Quantification of NEK7 levels in western blots as in (**b**), normalized to α-tubulin; n = 3 independent hippocampal cultures. *P < 0.05 by two-tailed *t*-test. **d** Mouse cortex homogenates obtained at the indicated stages were immunoblotted against NEK7 and α-tubulin as loading control. **e** Quantification of NEK7 levels in western blots as in (**d**), normalized to α-tubulin; n = 3 independent homogenates. *P < 0.05 by two-tailed *t*-test. **f** Lysates of somato-dendritic and axonal compartments of hippocampal neurons at 9 DIV were immunoblotted against NEK7, Tau, and pan-histone, using α-tubulin as loading control. **g** One DIV neurons infected with control and *Nek7* shRNA virus were lysed at 5 DIV and immunoblotted against NEK7. α-Tubulin served as loading control. **h** GFP-expressing control and NEK7-depleted neurons, transfected at 7 DIV were fixed at 14 DIV and stained with anti-GFP antibody. Scale bar, 50 µm. **i** Quantification of total dendrite length in control and NEK7-depleted neurons transfected at 1 DIV and analyzed at 5 DIV; n = 3 independent experiments. Total number of neurons analyzed: 90 (Control), 83 (shNek7#1), and 63 (shNek7#2); n.s. non-significant by two-tailed *t*-test. **j** Quantification of total dendrite length in 14 DIV neurons as in (**h**); n = 4 independent experiments. Total number of neurons: 35 (Control), 25 (shNek7 #1), and 28 (shNek7#2), *P < 0.05, **P < 0.01, ***P < 0.001 by two-tailed *t*-test. **k** Sholl analysis of 14 DIV neurons as in (**h**), schematic representation on the right. Mean number of intersections are plotted. Statistics as in (**j**). Error bars: s.e.m. Columns in all graphs show means and dot overlays individual data points

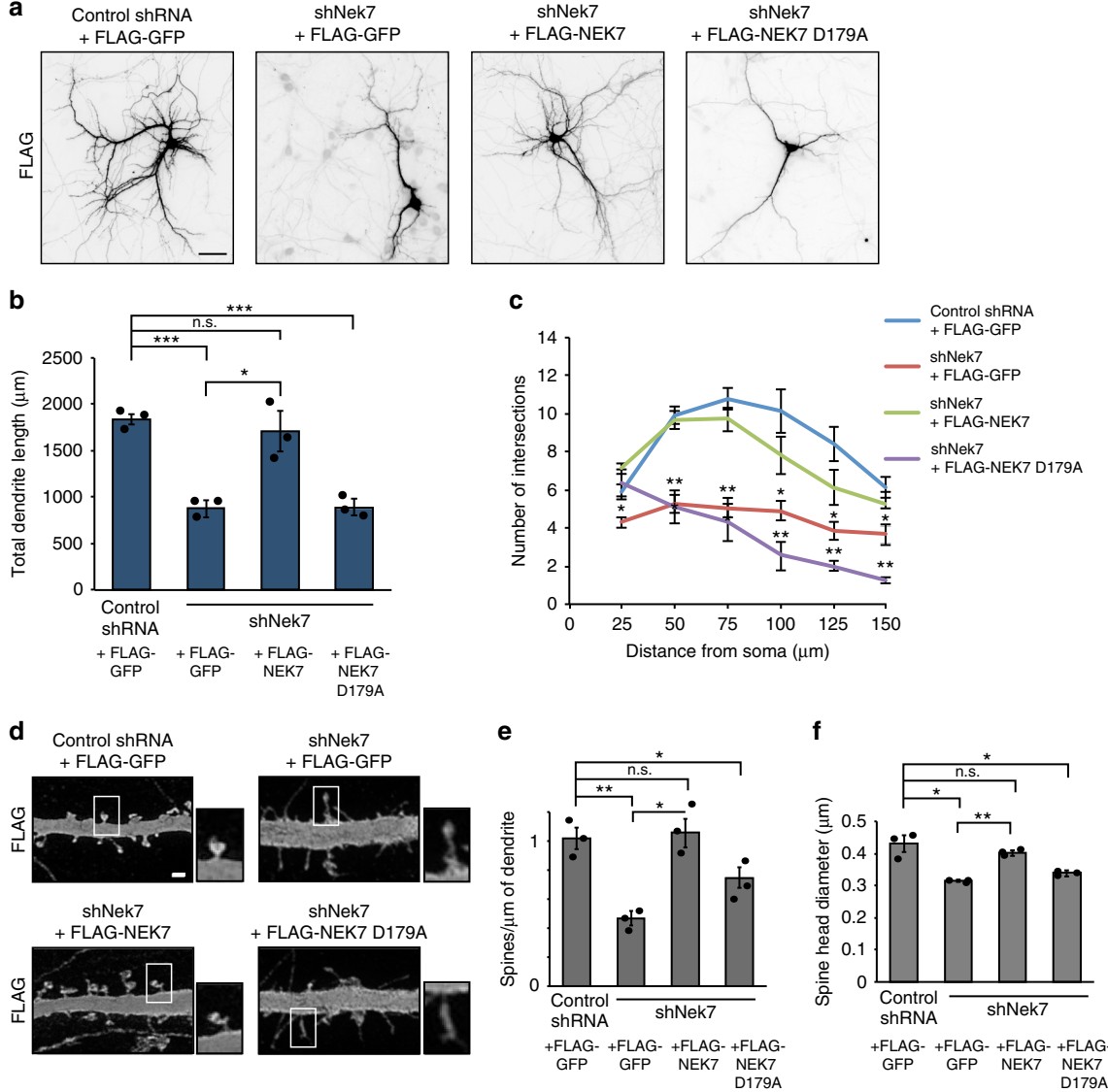

**Fig. 2** NEK7 kinase activity is required for dendrite and spine morphogenesis. **a** Fourteen DIV neurons, co-transfected at 7 DIV with control or *Nek7* shRNA and with plasmid expressing FLAG-tagged GFP, NEK7, or NEK7 D179A kinase-dead mutant were fixed and stained with anti-FLAG antibody. Scale bar, 50 μm. **b** Quantification of the total dendrite length in 14 DIV neurons in the conditions shown in (**a**); $n = 3$ independent experiments. Total number of neurons: 42 (Control), 33 (Nek7 depleted), 28 (rescue with WT Nek7), and 30 (rescue with Nek7 D179A). $*P < 0.05$, $**P < 0.01$, $***P < 0.001$, by two-tailed $t$-test. **c** Sholl analysis of 14 DIV neurons in the conditions shown in (**a**). Mean number of intersections are plotted. Statistics as in (**b**). **d** Representative confocal images of dendritic spines in primary dendrites of anti-FLAG-stained neurons as in (**a**). Scale bar, 1 μm. **e** Quantification of the density of spines as in (**d**); $n = 3$ independent experiments, total of 20 to 25 neurons per condition, total number of spines analyzed per condition $366 < n < 893$. n.s. non-significant, $*P < 0.05$, $**P < 0.01$ by two-tailed $t$-test. **f** Quantification of head diameter of spines as in (**d**). Statistics as in (**e**). Error bars: s.e.m.

branching (Supplementary Fig. 3b, d). While in Eg5-inhibited cells the overall spine density was reduced (0.93 spines/μm compared to 1.31 spines/μm in control cells), spine head morphology was not affected (Supplementary Fig. 3e–g). In summary, the defects in dendrite length and arborization after inhibition of Eg5 were similar to those caused by depletion of NEK7, but spine defects were less pronounced.

**An Eg5 phospho-mimetic mutant rescues NEK7 depletion defects**. In human mitotic cells NEK7 phosphorylates Eg5 at S1033[26]. We asked whether phosphorylation of Eg5 had any role in dendrite morphogenesis and could be linked to the NEK7-depletion phenotypes. Neurons at 7 DIV were infected with shRNA to deplete NEK7 and co-transfected with plasmid

expressing FLAG-EGFP, FLAG-tagged human wild-type Eg5, phospho-null (S1033A) Eg5 mutant, or phospho-mimetic (S1033D) Eg5 mutant (Fig. 3a). As expected, NEK7 shRNA decreased total dendrite length (1051 μm compared to 1884 μm in control cells) in neurons that co-expressed FLAG-EGFP as control (Fig. 3b, c). Strikingly, this defect was fully rescued by co-expressing the phospho-mimetic S1033D Eg5 mutant (2036 μm dendrite length), but not by wild-type Eg5 or the S1033A mutant (1160 μm and 1439 μm dendrite length, respectively) (Fig. 3b, c). Similarly, only the S1033D phospho-mimetic Eg5 mutant was able to rescue the dendrite branching defect (Fig. 3b, d). We also analyzed dendritic spines in the same conditions, and found that the decreased spine density in NEK7-depleted cells (0.50 spines/μm compared to 1.01 spines/μm in non-depleted cells) was only partially rescued by the

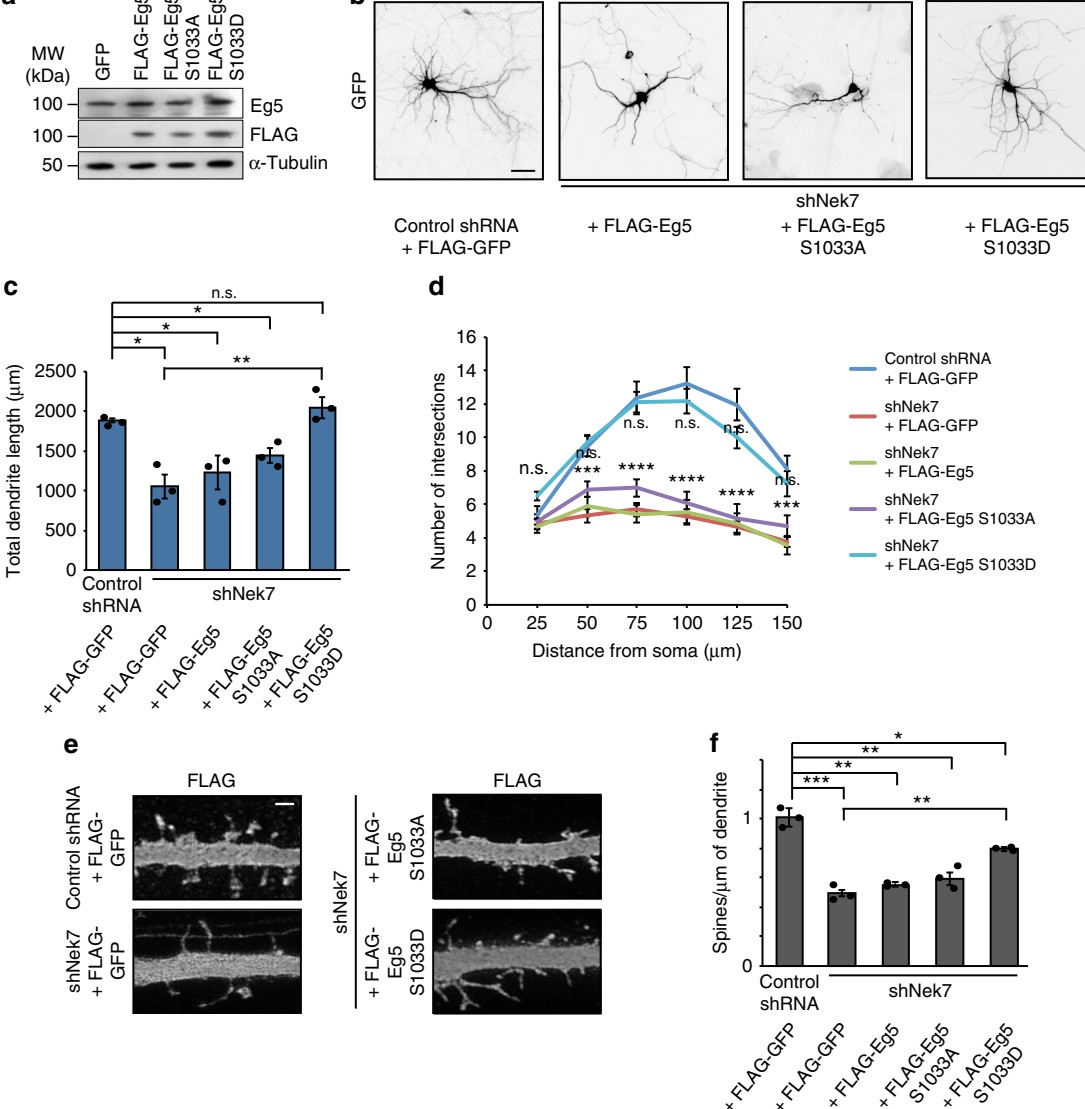

**Fig. 3** Expression of a phospho-mimetic mutant of Eg5 rescues defects in NEK7-depleted neurons. **a** Extracts from 14 DIV neurons infected at 7 DIV with lentivirus containing pLL3.7 plasmids that express either GFP, FLAG-Eg5 wild-type, or S1033 mutants. Lysates were probed with Eg5 and FLAG antibodies by western blotting. α-Tubulin was used as loading control. **b** Neurons, infected at 7 DIV with control or *Nek7* shRNA virus, and co-transfected with FLAG-GFP or FLAG-Eg5 (wild type or mutants as indicated) and GFP plasmid, were fixed at 14 DIV and co-stained with GFP antibody (shown in the figure) and with FLAG antibody (for selecting cells with similar levels of FLAG-Eg5 expression). Scale bar, 50 μm. **c** Quantification of total dendrite length in 14 DIV neurons as in (**b**). **d** Sholl analysis of 14 DIV neurons as in (**b**). Mean number of intersections are plotted; n = 3 independent experiments. Total number of neurons: 30 (shControl+FLAG-GFP), 33 (shNek7+FLAG-GFP), 27 (shNek7+FLAG-Eg5), 22 (shNek7+FLAG-Eg5 S1033A), and 24 (shNek7+FLAG-Eg5 S1033D); n.s. non-significant, *P < 0.05, **P < 0.01, ***P < 0.001, ****P < 0.0001 by two-tailed *t*-test. **e** Representative images of dendritic spines in primary dendrites of anti-GFP-stained neurons as in (**b**). Scale bar, 1 μm. **f** Quantification of the density of dendritic spines as in (**d**); n = 3 independent experiments, total of 11 to 17 neurons per condition, total number of spines analyzed per condition 292 < n < 667. *P < 0.05, **P < 0.01, ***P < 0.001 by two-tailed *t*-test. Error bars: s.e.m. Columns in all graphs show means and dot overlays individual data points

S1033D mutant (0.80 spines/μm), while co-expression of wild-type Eg5 or the S1033A Eg5 mutant had no effect (0.54 and 0.59 spines/μm, respectively) (Fig. 3e, f). Together, these results support the model that NEK7 controls dendrite morphogenesis through phosphorylation of Eg5.

**NEK7 promotes Eg5 accumulation in distal dendrites**. In mitotic prophase NEK7-dependent S1033 phosphorylation promotes Eg5 accumulation on microtubules around centrosomes[26]. To investigate whether phosphorylation also affected Eg5 distribution in neurons, we visualized endogenous Eg5 by

immunofluorescence microscopy in dendrites of control, NEK7-depleted, and NEK7-depleted/rescued neurons at 14 DIV. The ability of the Eg5 antibody to label endogenous mouse Eg5 was confirmed by immunofluorescence analysis of mouse embryonic fibroblasts, which revealed the well-established Eg5 localization around the centrosomes in prophase, and at spindle poles and along spindle microtubules in prometaphase/metaphase as previously described in cycling cells[38] (Supplementary Fig. 4a, b). Moreover, the cytoplasmic Eg5 staining that we observed in neurons was reduced in cells expressing *Eg5* shRNA, indicating the specificity of the staining (Supplementary Fig. 4c–g). In neurons expressing control shRNA and FLAG-EGFP endogenous

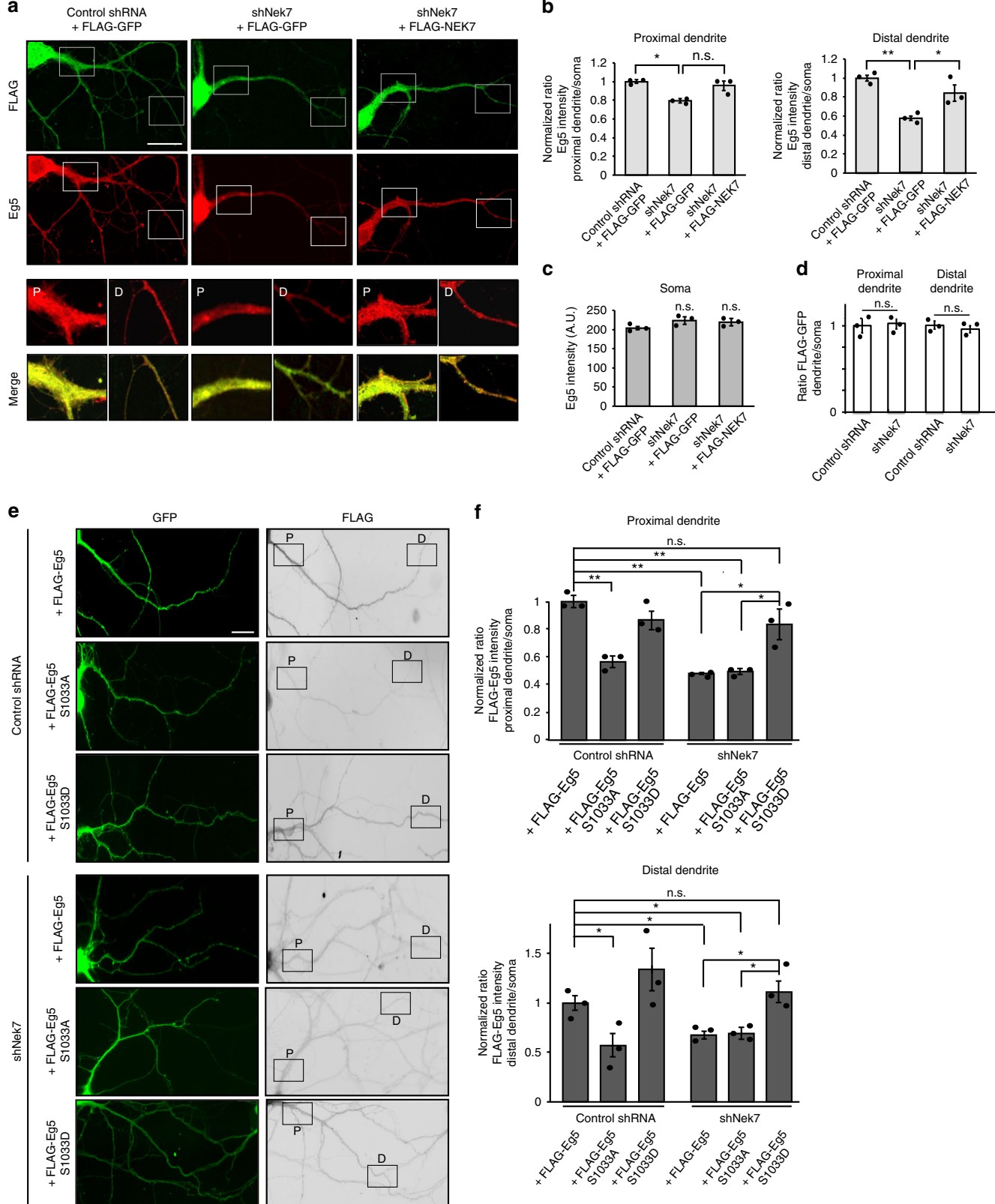

Eg5 was distributed throughout the dendrites (Fig. 4a, b). In NEK7-depleted cells, Eg5 intensity in dendrites (measured as ratio dendrite intensity/soma intensity) was slightly reduced in the proximal part and more strongly reduced in the distal part (~20% and ~40% decrease, respectively). This phenotype was not caused by changes in the total levels of Eg5, which remained unaltered (Supplementary Fig. 4c), and could be partially rescued by expression of wild-type FLAG-NEK7 (Fig. 4a, b). The intensities of Eg5 in the soma and in different parts of the axon were not significantly affected in any of the conditions (Fig. 4c, Supplementary Fig. 4h, i). As additional control we also quantified the distribution of cytosolic FLAG-GFP, which was unaffected by

**Fig. 4** NEK7 promotes Eg5 accumulation in distal dendrites through S1033 phosphorylation. **a** Neurons transfected with the indicated plasmids at 7 DIV were fixed and stained with FLAG and Eg5 antibodies. Magnifications show Eg5 staining in proximal (P) and distal (D) dendrite regions. Scale bar, 50 μm. **b** Quantification of the Eg5 intensity in ~50 μm segments of proximal and distal dendrites of neurons as in (**a**); $n = 3$ independent experiments (total number of neurons: 27 (Control), 24 (Nek7 depleted), and 23 (rescue with WT Nek7), 2 to 6 dendrites analyzed per neuron); n.s. non-significant, *$P < 0.05$, **$P < 0.01$, by two-tailed $t$-test. **c** Eg5 intensity in somas of neurons as in (**a**). Statistics as in (**b**). **d** FLAG staining in the same neurons as in (**a**) was quantified as control. Intensities were plotted relative to the intensities in the somas of the same cells. Statistics as in (**b**). **e** Seven DIV neurons were infected with control or *Nek7* shRNA virus, and co-transfected with FLAG-Eg5 (wild-type or S1033 mutants) and GFP plasmids, and fixed at 14 DIV. Images show dendrites after staining with GFP and FLAG antibodies. Scale bar, 25 μm. **f** Quantification of FLAG-Eg5 intensity in ~50 μm segments of proximal and distal regions of dendrites in 14 DIV neurons as in (**e**), normalized to GFP intensity and to FLAG-Eg5 intensities in the somas of the same cells; $n = 3$ independent experiments (total number of neurons: 20 to 22 per condition, 2 to 5 dendrites measured per neuron); n.s. non-significant, *$P < 0.05$, **$P < 0.01$ by two-tailed $t$-test. Error bars: s.e.m. Columns in all graphs show means and dot overlays individual data points

depletion of NEK7 (Fig. 4d). We conclude that NEK7 controls Eg5 distribution in dendrites, promoting its accumulation in the distal parts.

To explore whether the NEK7-dependent accumulation of Eg5 in distal dendrites was linked to the S1033 phosphorylation state, we examined the distribution of transfected FLAG-tagged wild-type, S1033A mutant, and S1033D mutant Eg5 in control and NEK7-depleted neurons at 14 DIV. Co-transfection of EGFP-expressing plasmid allowed us to trace transfected cells and to normalize FLAG-Eg5 intensity to compensate for cell-to-cell variability in the overall expression levels. In control shRNA-transduced neurons, FLAG-Eg5 wild-type and S1033D mutant intensity values were similar in the proximal and distal parts of dendrites, displaying a distribution that was similar to endogenous Eg5 (Fig. 4e, f). For the S1033A mutant, however, we measured reduced intensities in both proximal and distal dendrite regions (~30% and ~40% decrease, respectively). In NEK7-depleted neurons the intensities of FLAG-tagged wild-type Eg5 and S1033A mutant Eg5 were decreased throughout the entire dendrite when compared to their intensities in non-depleted neurons. Remarkably, in NEK7-depleted neurons only the S1033D mutant was able to accumulate in both proximal and distal dendrite areas, with intensity values similar to those in non-depleted neurons (Fig. 4e, f). Taken together, these observations indicate that NEK7, through phosphorylation of S1033, promotes localization of Eg5 to proximal and distal parts of the dendrites.

**Eg5 dendritic accumulation limits minus-end-out microtubules.** To learn more about the mechanism by which NEK7 and Eg5 regulate dendrite length and arborization, we studied how Eg5 affects microtubule organization in this compartment. Previous work showed that Eg5 inhibition by monastrol altered microtubule polarity in dendrites by increasing the percentage of retrogradely growing microtubules[36]. We tested whether knockdown of NEK7 had similar effects. We depleted NEK7 by lentiviral transduction at 3 DIV and transiently transfected these cells at 8 DIV with plasmid expressing EB3-tomato to label growing microtubule plus ends. We found that in NEK7-depleted cells the density of EB3 comets in the distal part of dendrites was slightly increased when compared to control cells (Fig. 5a, b) and that, intriguingly, the percentage of retrograde comets was also increased (22.0% compared to 14.8% in controls) (Fig. 5c; arrowheads in Fig. 5a; Supplementary Movie 1). We did not observe any changes in comet number or polarity in axons, suggesting that NEK7 specifically regulates dendritic microtubules (Supplementary Fig. 5a–c). We also analyzed EB3-tomato comets in neurons after inhibition of Eg5. Similar to NEK7 depletion, STLC treatment increased EB3 comet density (Fig. 5d, e) and increased the percentage of retrograde comets in the distal part of dendrites (22.5% compared to 17.7% in controls) (Fig. 5f; arrowheads in Fig. 5d). The similarity of the phenotypes strongly suggests that the NEK7-dependent accumulation of Eg5 in

dendrites controls microtubule polarity in this compartment. We wondered whether altered microtubule polarity per se was sufficient to cause the reduction in dendrite length and arborization that we had observed in NEK7/Eg5-inhibited neurons. In a previous study we showed that overexpression of the small CDK5RAP2 51–100 fragment, which stimulates γTuRC to ectopically nucleate microtubules[39], increased retrograde microtubule growth in axons[15]. We tested whether this approach may also be used to alter microtubule polarity in dendrites. We transiently transfected neurons at 6 DIV with the CDK5RAP2 51–100 fragment fused to GFP, together with the EB3-Tomato reporter. Indeed, analysis of EB3 comets at 9 DIV revealed an increase in comet density in both proximal and distal dendrite regions (Fig. 5g, h). Importantly, the percentage of retrograde comets in distal dendrite regions was also increased (22.8% compared to 13.0% in controls) (Fig. 5g, i). Moreover, similar to neurons depleted of NEK7 or treated with Eg5 inhibitor, neurons overexpressing EGFP-CDK5RAP2 51–100 had shorter and less branched dendrites than control cells expressing EGFP alone (Fig. 5j–l). Together, these results suggest that increasing retrograde microtubule growth in the distal part of dendrites is sufficient to impair dendrite morphogenesis.

**Eg5 regulates dendrite microtubule polarity by stabilization.** How does Eg5 regulate microtubule polarity in dendrites? Eg5 "loop-5" inhibitors such as monastrol or STLC not only inhibit motor activity by stabilizing bound nucleotide, but also weaken Eg5 interaction with microtubules. In contrast, "rigor" inhibitors such as FCPT (2-(1-(4-fluorophenyl)cyclopropyl)-4-(pyridin-4-yl)thiazole) lock Eg5 in a microtubule-bound state[40–42]. Indeed, treatment of neurons with STLC decreased Eg5 intensity in both proximal and distal dendrites, whereas FCPT had the opposite effect (Supplementary Fig. 5d, e). Strikingly, the two inhibitors also differentially affected microtubule polarity. Rather than increasing the percentage of retrogradely growing microtubules as seen for STLC, FCPT had the opposite effect and decreased the percentage of retrograde EB3 comets (11.8% compared to 16.5% in controls) (Fig. 6a, b). This result suggested that retrograde microtubule growth in dendrites is negatively regulated by the binding of Eg5 to microtubules, independent of its motor activity.

The above findings would be in agreement with the previously proposed model that distinct microtubule polarities in axon and dendrites are established and maintained by motor-driven sorting of short microtubules, and that Eg5 functions as a brake on the transport of minus-end-out microtubules into dendrites[17,43–45]. However, live imaging of microtubule bleach marks in cultured hippocampal neurons expressing EOS-α-tubulin did not reveal any transport or sliding of microtubules in dendrites (Supplementary Movie 2). Considering that in dendrites minus-end-out microtubules are less dynamic than plus-end-out microtubules[4], an increase in retrograde comets may also be explained by an increased dynamicity of minus-end-out microtubules, resulting

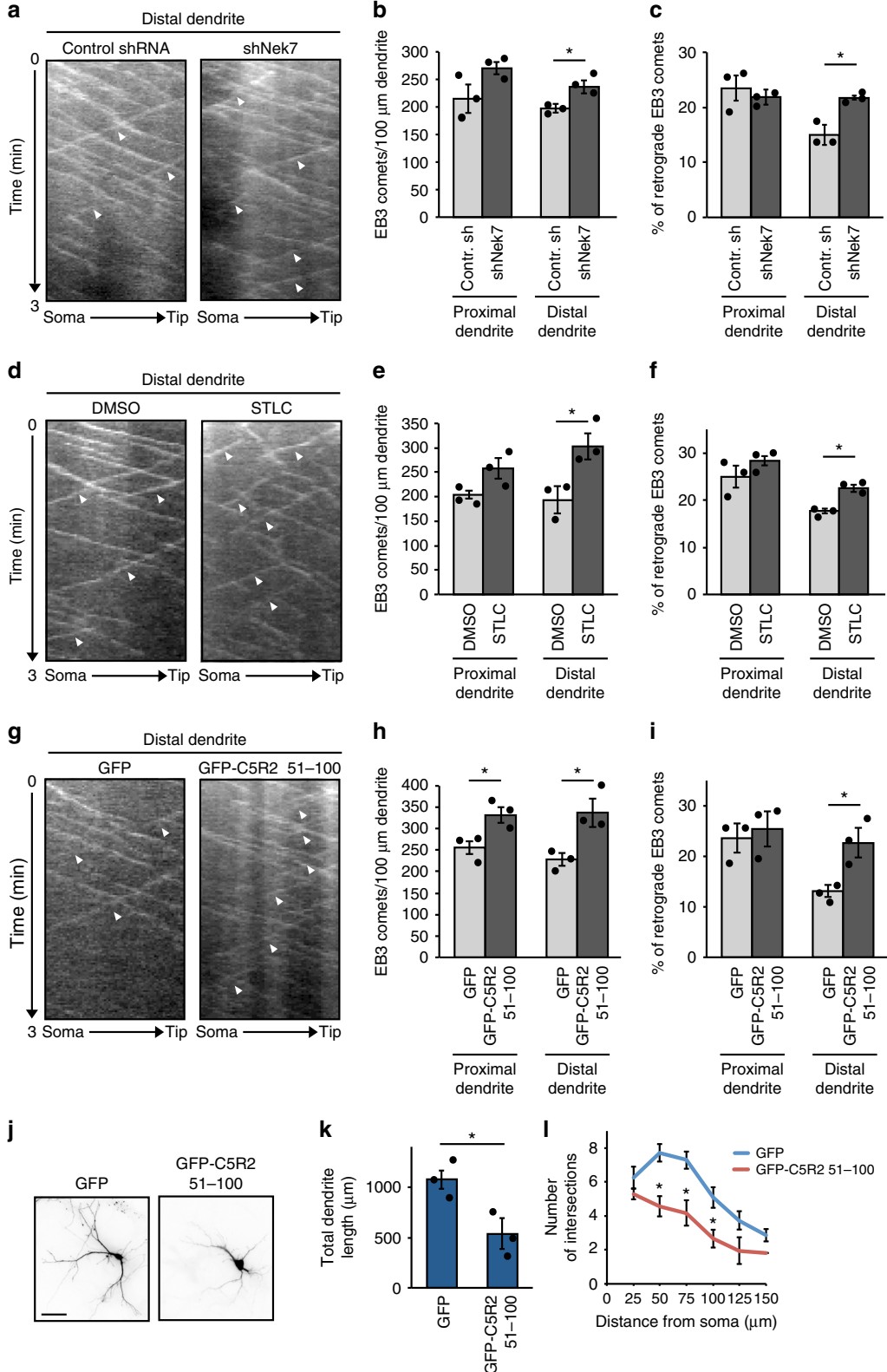

from destabilization. In this model, rather than opposing the transport of short microtubules into dendrites, Eg5 would function mainly as microtubule stabilizer. To test whether Eg5 binding may stabilize dendritic microtubules we treated neurons with STLC or FCPT and stained and quantified the proportion of microtubules modified by acetylation, a modification typically enriched on stable microtubules. Consistent with our hypothesis,

displacing Eg5 from microtubules with STLC reduced the amount of acetylated microtubules, whereas locking Eg5 in the microtubule-bound state with FCPT increased the signal (Fig. 6c, d). Since the localization of Eg5 to dendrites is promoted by NEK7, we predicted that NEK7 depletion would also destabilize dendritic microtubules. Indeed, this is what we observed (Supplementary Fig. 5g, h). Previous work suggested that

**Fig. 5** NEK7 and Eg5 limit retrograde microtubule growth in distal dendrites. **a** Hippocampal cultures were infected with control or *Nek7* shRNA virus at 3 DIV, transfected with EB3-tomato at 8 DIV, and analyzed by time-lapse microscopy at 9 DIV. Kymographs show EB3-tomato comet dynamics in distal dendrites. White arrowheads indicate retrograde EB3 comets. **b** Quantification of EB3 comet density in proximal and distal segments of dendrites in control or Nek7-depleted neurons; n = 3 independent experiments. Total of 29 (Control shRNA) and 26 (shNek7) dendrites. *P < 0.05 by two-tailed *t*-test. **c** Quantification of the polarity of comets analyzed in (**b**). Statistics as in (**b**). **d** Neurons were treated with DMSO or STLC at 5 DIV and transfected with EB3-tomato reporter and analyzed as in (**a**). Kymographs show EB3-tomato comet dynamics in distal dendrites. White arrowheads indicate retrograde EB3 comets. **e** Quantification of EB3 comet density in proximal and distal segments of dendrites in DMSO or STLC-treated neurons; n = 3 independent experiments. Total of 29 (DMSO) and 27 (STLC) dendrites. *P < 0.05 by two-tailed *t*-test. **f** Quantification of the polarity of EB3 comets in (**e**). Statistics as in (**e**). **g** Hippocampal cultures were co-transfected with GFP or GFP-CDK5RAP2 51–100, and the reporter EB3-tomato at 6 DIV, and analyzed by time-lapse microscopy at 9 DIV. Kymographs show EB3-tomato comet dynamics in distal dendrites. White arrowheads indicate retrograde EB3 comets. **h** Quantification of EB3 comet density in proximal and distal segments of dendrites in neurons expressing GFP or GFP-CDK5RAP2 51–100; n = 3 independent experiments. Total of 24 dendrites (GFP) and 25 dendrites (GFP-C5R2 51–100). *P < 0.05 by two-tailed *t*-test. **i** Quantification of EB3 comet polarity in the segments analyzed in (**h**). Statistics as in (**h**). **j** Nine DIV neurons expressing GFP or GFP-CDK5RAP2 51–100 were stained with GFP antibody. Scale bar, 25 μm. **k** Quantification of total dendrite length of neurons as in (**j**); n = 3 independent experiments, total number of neurons: 71 (GFP) and 36 (GFP-CDK5RAP2 51–100). *P < 0.05 by two-tailed *t*-test. **l** Mean number of dendrite intersections in neurons as in (**j**). Statistics as in (**k**). Error bars: s.e.m. Columns in all graphs show means and dot overlays individual data points

acetylated (presumably more stable) microtubules are less abundant in distal compared to proximal dendrites[46]. This would explain why NEK7 depletion or Eg5 inhibition with STLC, treatments that destabilize microtubules, would have stronger effects in the distal region where microtubules have an overall lower stability. Indeed, we could also confirm this reduction in acetylated microtubules in distal dendrites in our experimental condition (Supplementary Fig. 4j, k).

To have a stabilizing effect we would predict that Eg5 is also relatively stably bound to dendritic microtubules. To test this we analyzed the Eg5–microtubule interaction in neurons by photo-conversion of transfected Eg5-EOS (Fig. 7a–c). Indeed, whereas photoconverted EOS alone rapidly diffused away from the area targeted by the laser, a fraction of photoconverted Eg5-EOS remained stationary in the photoconverted region during 4 min, suggesting stable interaction of Eg5-EOS with dendritic microtubules in the absence of significant lateral movement of either component. Consistently, photoconversion of EOS-α-tubulin also showed no lateral movement of the fluorescent mark (Fig. 7a–c). When the experiment was performed in the presence of STLC the Eg5-EOS signal disappeared rapidly from the converted region, consistent with the notion that STLC weakens Eg5 interaction with microtubules (Fig. 7a–c; Supplementary Movie 3).

We then asked whether treatment with FCPT, which promotes Eg5 binding and stabilization of microtubules, in the absence of motor activity, was sufficient for rescuing dendrite defects in NEK7-depleted neurons. While FCPT treatment only very mildly rescued dendrite length, spine density was fully restored. Spine morphology defects, however, were not rescued (Supplementary Fig. 6a–e). These results suggest that microtubule stabilization is an important part of Eg5 function, but the ATP-dependent motor activity is also required.

Having established NEK7 as a regulator of dendrite morphogenesis in cultured neurons, we finally sought to determine whether this regulation also occurred in vivo. To address this we obtained mice carrying a gene trap (gt) insertion between exons 5 and 6 of the *Nek7* gene. Western blotting of e18.5 whole embryo homogenates confirmed the absence of NEK7 protein in *Nek7*^gt/gt^ animals (Supplementary Fig. 6f, g). First, we analyzed hippocampal neurons prepared from *Nek7*^gt/gt^ embryos in culture. At 9 DIV, *Nek7*^gt/gt^ neurons displayed, similar to NEK7 knockdown neurons, an increase in both EB3 comet density and in the percentage of retrograde comets in distal dendrites. (Fig. 8g–i). Interestingly, no significant changes in overall dendrite length were observed at this stage (Supplementary Fig. 6k, l). However, at 14 DIV neurons from *Nek7*^gt/gt^ embryos had shorter and less branched dendrites compared to neurons from WT littermates (Fig. 8a–c) and

displayed a reduction in spine density and spine head diameter (Fig. 8d–f). In addition, dendritic Eg5 staining was reduced, in particular in the more distal part (Supplementary Fig. 6h–j). Taken together, these findings confirm the results obtained with NEK7 knockdown cultures and additionally reveal that the alterations in dendritic microtubule organization precede the morphological dendrite and spine defects. In agreement with previous reports[47,48] *Nek7*^gt/gt^ animals from crosses of heterozygous parents were born at sub-Mendelian ratio. At birth, *Nek7*^gt/gt^ mice were similar in size and appearance when compared to wild-type or heterozygous littermates. At 3–4 months of age, however, *Nek7*^gt/gt^ animals developed limb paresis and displayed up to 20% weight loss (data not shown), similar to what was described for independently generated *Nek7*-deficient mice in a previous study[47].

To determine if neurons in *Nek7*^gt/gt^ brains displayed dendritic defects, we used a gene gun for diolistic labeling of individual neurons in brain sections with a diffusible dye. We then determined total dendrite length and extent of branching as well as the density and morphology of spines for pyramidal neurons in the CA1 hippocampal region. We observed that neurons in *Nek7*^gt/gt^ brains had shorter and less branched basal dendrites (Fig. 8j–l) and that the density and head diameter of dendritic spines were reduced compared to wild-type controls (Fig. 8m–o). These results indicate that NEK7 is required for correct dendrite and spine formation in vivo.

## Discussion

In this work we have explored the concept that key components of the microtubule cytoskeleton in cycling cells, including proteins involved in mitotic spindle assembly, may also have important roles in postmitotic, differentiated cells. This "recycling" concept has been proposed for neurons almost two decades ago[49], but has never been explored systematically.

As a first step towards this goal we have performed micro-array analysis of cultured mouse hippocampal neurons from samples taken over the course of 2 weeks (0 DIV–15 DIV) during neuron differentiation and maturation. This allowed us to generate expression profiles for all of the genes known to be involved in microtubule organization in cycling and mitotic cells. Supporting the hypothesis that many microtubule regulators used by proliferating cells are recycled in postmitotic cells, we found that among the differentially expressed genes only about 46% are downregulated, whereas another 48% are upregulated. Genes in this latter group, together with the remaining genes that showed transient changes in expression with no overall up- or down regulation, are the most interesting

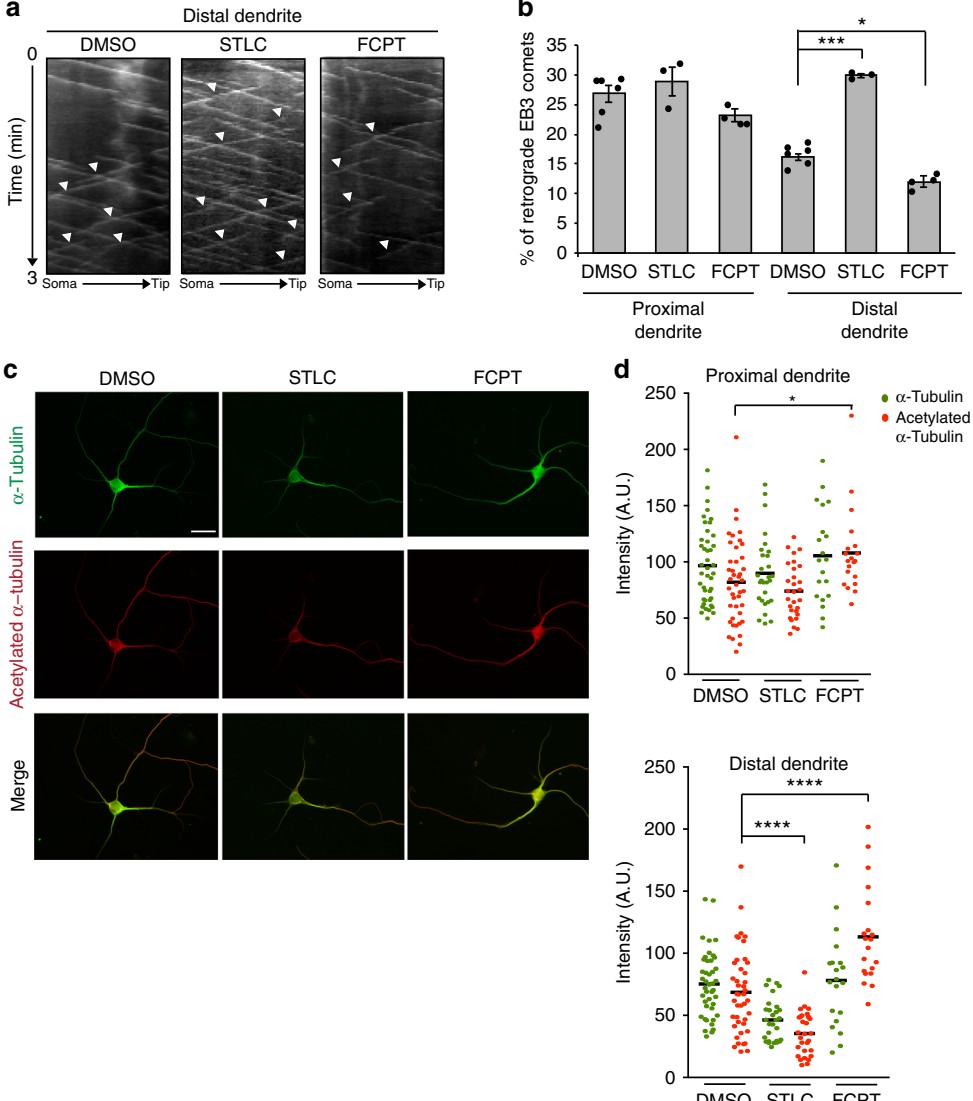

**Fig. 6** Eg5 limits retrograde microtubule growth in dendrites by microtubule stabilization. **a** Kymographs revealing EB3 comet dynamics in distal dendrites of DMSO-, STLC-, or FCPT-treated neurons. Neurons were treated with the drugs at 3 DIV, transfected with the reporter EB3-tomato at 8 DIV, and analyzed by time-lapse microscopy at 9 DIV. White arrowheads indicate retrograde EB3 comets. **b** Quantification of EB3 comet polarity in proximal and distal segments of dendrites in neurons as in (**a**); $n = 3$ to 6 independent experiments. Columns show means and dot overlays individual data points. Total of 29 (DMSO), 29 (STLC), and 30 (FCPT) dendrites. *$P < 0.05$, ***$P < 0.001$ by two-tailed $t$-test. Error bars: s.e.m. **c** Neurons treated as in (**a**) were fixed and stained with antibodies against α-tubulin and acetylated α-tubulin as indicated. Scale bar, 25 μm. **d** Quantification of α-tubulin (green dots) and acetylated α-tubulin (red dots) staining intensity in proximal and distal dendrite regions of neurons as in (**c**). Each dot represents the average value for one cell (2–4 dendrites); $n = 43$ neurons (DMSO), 29 (STLC), and 20 (FCPT) from 3 independent experiments. Black horizontal line: mean of the three experiments. *$P < 0.05$, ****$P < 0.0001$, by two-tailed $t$-test

candidates for further analysis in future studies. Indeed, validating the approach that we took here, during our study two genes in this group, *cyclin E1/Ccne1* and *Mtap7*, were shown to have roles in neurons[50–53].

Apart from differentially expressed genes, a large number of genes did not display expression changes during the sampling time course. While we did not analyze these genes further, it should be noted that, despite the absence of differential expression, some may be expressed throughout neuron differentiation and maturation and thus could have important roles. Again, this was confirmed by work published during the course of this study, revealing roles for *Ckap5/ch-TOG* and *Tacc3* in postmitotic neurons[54–56].

One of the genes displaying strong upregulation during neuron differentiation is *Nek7*, which encodes a member of the NIMA-

related kinase (NEK) family that has previously been implicated in mitosis and cell division[24–27,48]. More recently, NEK7 was shown to also have a kinase-independent role as essential component of the inflammasome, a multi-protein complex of the innate immune system[47,57,58].

Here, we reveal a novel role of NEK7 in regulating dendrite and spine morphogenesis in postmitotic neurons. Interestingly, NEK6, which is highly similar to NEK7 (~85% identical in their catalytical domains) and NEK9, a NEK7 activator in mitosis[26,33,59], do not seem to contribute to these functions. While the mitotic regulatory circuit with NEK6 and NEK9 is not maintained, the downstream modulation of a kinesin by NEK7 is: similar to cycling cells, where NEK7 controls prophase centrosome separation and cytokinesis through two kinesin motors (Eg5 and KIF14, respectively)[26,27], NEK7 role in dendrites is in

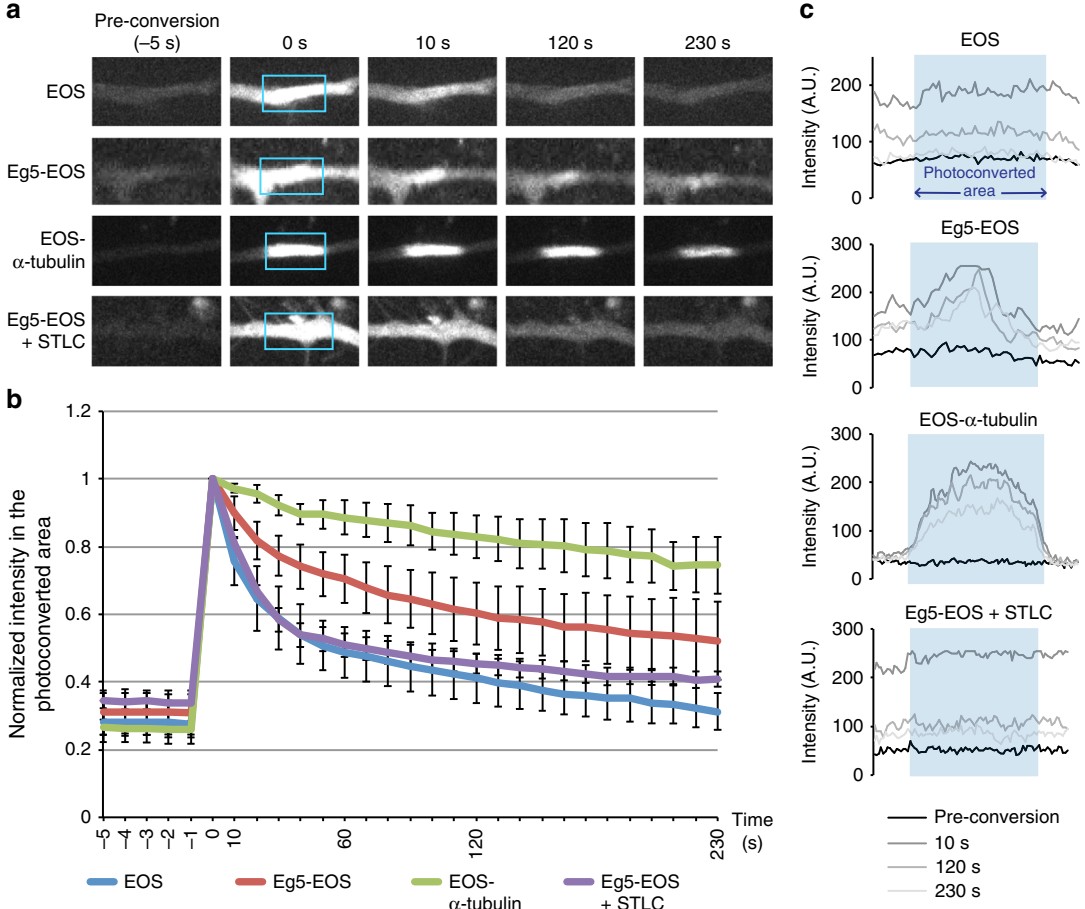

**Fig. 7** A fraction of EOS-tagged Eg5 in dendrites is stably bound to microtubules. **a** Time-lapse image series showing photoconverted EOS signal in dendrites of 9 DIV neurons expressing EOS, EOS-tagged Eg5, or EOS-tagged α-tubulin. Blue boxes indicate the photoconverted region at time point 0. **b** Increase and decay of the photoconverted EOS signal intensity over time in the targeted area of dendrites as in (**a**). For each sample signal intensities were normalized to the intensity at the time point of photoconversion, which was set to 1. For each time point mean values from three different experiment were plotted; $n = 3$ experiments, total of 9–19 neurons per condition. Error bars: s.e.m. **c** Fluorescence intensity distributions of photoconverted EOS at four different time points along photoconverted and adjacent regions in dendrites shown in (**a**). Blue shading delimits the section that was photoconverted

part mediated by Eg5. Moreover, similar to the NEK7-dependent recruitment of Eg5 around centrosomes at prophase[26], our results show that in neurons modification of a phosphorylation site in the Eg5 C-terminal domain (S1033 in human Eg5 or an ortho-logous residue in mice) promotes recruitment of the kinesin to microtubules in distal dendrite regions, suggesting a similar regulation of Eg5 localization in both systems.

While Eg5 inhibition using RNAi or monastrol was shown to reduce dendrite length previously[36], our experiments also impli-cate NEK7–Eg5 in the formation of dendritic spines, consistent with our observation that NEK7 was enriched in the somato-dendritic compartment and upregulated during the later stages of differentiation, when neurons establish synaptic connections. In NEK7-depleted neurons spine density was reduced and spines were thin and filopodia-like and rarely displayed mushroom shape. However, while expression of the Eg5 S1033D mutant, which mimics NEK7-dependent phosphorylation, fully rescued reduced dendrite length after NEK7 RNAi, spine defects were only partially rescued. Moreover, chemical inhibition of Eg5 reduced spine density but did not affect spine morphology. These results suggest that regulation of spine formation and morphol-ogy may involve additional NEK7 phosphorylation sites and/or substrates. Several other kinesins, dynein, and non-motor microtubule-associated proteins have been implicated in various aspects of dendrite and/or spine morphogenesis[22,36,43,60–62].

Destabilization of the dendritic microtubule array has also been correlated with impaired spine formation[63]. How these different activities are coordinated is still unclear. Future work will show whether any of these effector molecules may also be regulated by NEK7.

How does NEK7-dependent recruitment of Eg5 to dendritic microtubules control dendrite length? Whereas at later stages of neuron development the percentage of minus-end-out micro-tubules in dendrites approximates the percentage of plus-end-out orientations, dendrites in young, developing neurons contain more plus-end-out microtubules, in particular when considering dynamic microtubules[4]. This increased percentage of plus-end-out microtubules may be important for dendrite growth and arborization. Indeed, increasing the proportion of retrogradely growing microtubules through a NEK7-independent mechanism, by overexpression of the nucleation promoting CM1 domain of CDK5RAP2, similarly impaired dendrite growth and branching. Taken together, the data suggest that Eg5 contributes to dendrite growth and branching by limiting the occurrence of retrogradely growing microtubules, in particular in distal dendrite regions. This observation would be consistent with the previously pro-posed function of Eg5 as a brake on the transport of short "minus-end-out" microtubules into dendrites by other motors[36]. Similarly, Eg5 was recently proposed to function as a "frictional brake" on microtubule movements during mitotic spindle

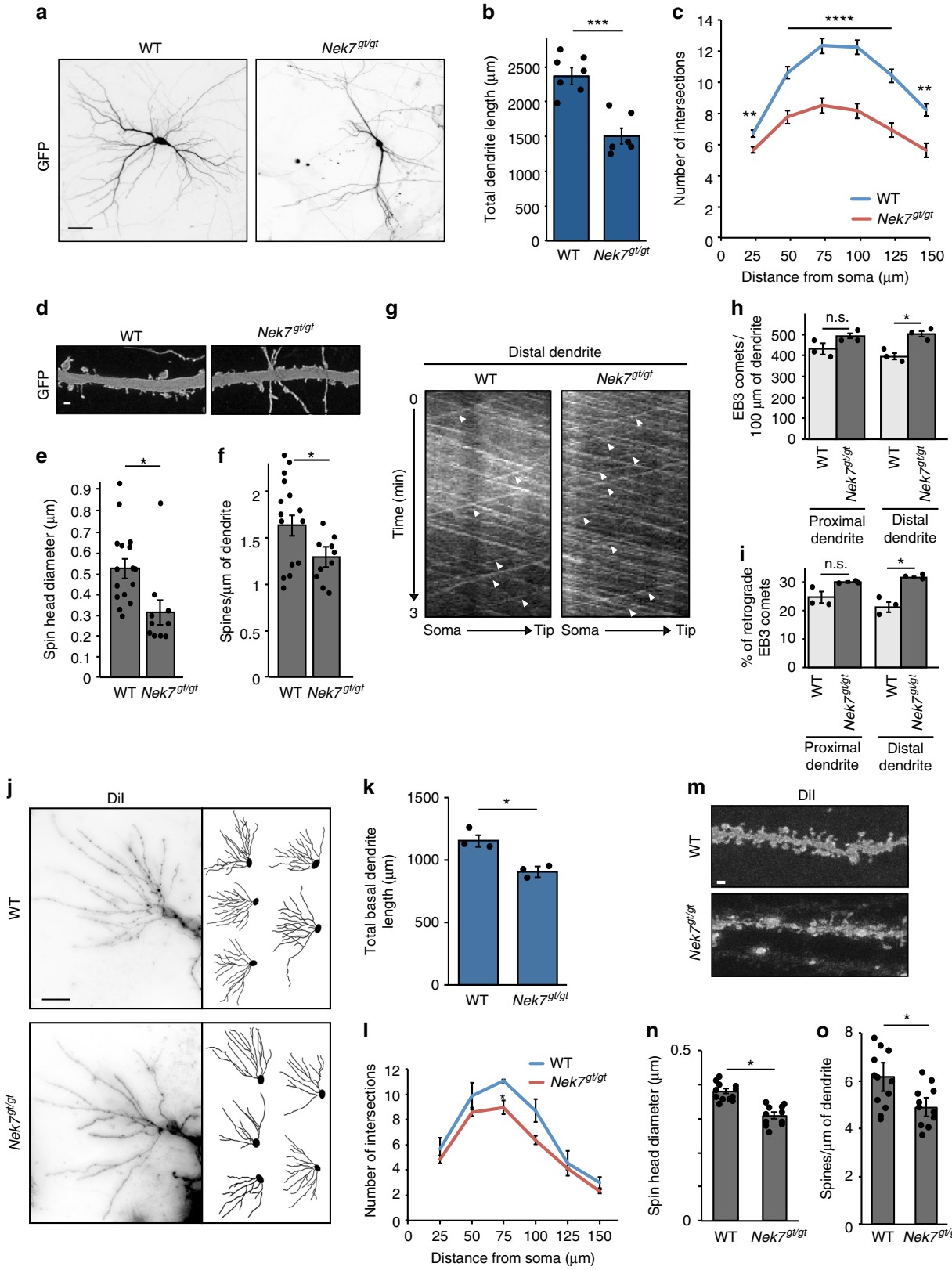

assembly to restrict spindle elongation at anaphase B[42]. However, we did not observe any transport of microtubules in dendrites. While it is possible that these transport events are relatively rare and thus were not detected under our experimental conditions, our finding that NEK7/Eg5 inhibition destabilizes dendritic microtubules provides an alternative explanation: destabilization

may result in an overall increase in microtubule dynamicity, which would be observed as an increase in the percentage of retrograde EB3 comets, since in dendrites microtubules with this orientation (minus-end-out) are normally less dynamic[4]. Similarly, an overall increase in dynamic microtubules may also explain the increase in retrograde comets that we observed after

**Fig. 8** Dendrite defects in hippocampal neurons of $Nek7^{gt/gt}$ mice. **a** GFP-expressing 14 DIV hippocampal wild-type and $Nek7^{gt/gt}$ neurons. Scale bar, 50 µm. **b** Total dendrite length in neurons as in (**a**); $n = 8$ wild-type and 7 $Nek7^{gt/gt}$ embryos. Total number of neurons: 139 (wild type) and 84 ($Nek7^{gt/gt}$). ***$P < 0.001$, by two-tailed $t$-test. **c** Mean number of intersections by Sholl analysis of neurons as in (**a**); $n = 126$ wild-type and 71 $Nek7^{gt/gt}$ neurons. **$P < 0.01$, ****$P < 0.0001$ by two-tailed $t$-test. **d** Spines in primary dendrites of neurons as in (**a**). Scale bar, 1 µm. **e** Density of spines as in (**d**); $n = 15$ wild-type and 10 $Nek7^{gt/gt}$ neurons, 3 embryos per genotype, 552 and 260 spines, respectively. *$P < 0.05$, by two-tailed $t$-test. **f** Head diameter of spines as in (**d**). Statistics as in (**d**). **g** Kymographs of EB3 comets in dendrites of wild-type or $Nek7^{gt/gt}$ 9 DIV neurons. White arrowheads indicate retrograde growth. **h** Comet density in dendrites of wild-type or $Nek7^{gt/gt}$ neurons. **i** Percentage of retrograde comets in the segments analyzed in (**h**). **h**, **i** $n = 3$ wild-type and 3 $Nek7^{gt/gt}$ embryos, 26 and 25 dendrites, respectively. n.s. non-significant, *$P < 0.05$ by two-tailed $t$-test. **j** Maximum intensity projections of diolistically labeled dendritic trees of CA1 wild-type or $Nek7^{gt/gt}$ neurons in 4-month-old mice and representative tracings. Scale bar, 25 µm. **k** Total length of dendrites shown in (**j**); $n = 3$ wild-type and 3 $Nek7^{gt/gt}$ mice (2 males and 1 female each), 54 neurons per genotype. *$P < 0.05$ by two-tailed $t$-test. **l** Mean intersection number in dendrite trees shown in (**j**). Number of animals as in (**j**), 41 and 44 neurons per genotype, respectively. *$P < 0.05$ by two-tailed $t$-test. **m** Dendritic spines in primary dendrites of neurons as in (**j**). Scale bar, 1 µm. **n** Head diameter of spines as in (**m**). **o** Density of spines as in (**m**); $n = 12$ wild-type and 11 $Nek7^{gt/gt}$ neurons, 3 embryos per genotype, 1263 and 823 spines, respectively. *$P < 0.05$, by two-tailed $t$-test. Error bars: s.e.m. Columns in all graphs show means and dot overlays individual data points

induction of ectopic nucleation by expression of EGFP-CDK5RAP2 51–100.

We were surprised to find that Eg5 controls microtubule stability and polarity in dendrites through its microtubule binding function, independently of its motor activity. Still Eg5 motor activity is important: the Eg5 inhibitor FCPT, which promotes microtubule stability by locking Eg5 in its microtubule-bound state, was not able to fully rescue the NEK7 knockdown phenotypes. Thus, in contrast to the activity of other MAPs, Eg5 may use its ability to move along microtubules to focus its stabilizing activity in specific regions. This would provide an additional level of regulation that is not available to proteins with a simple on/off microtubule binding mode.

In this study, we have analyzed the role of NEK7 in pyramidal neurons in the hippocampus, but it is likely that NEK7 has similar roles in other types of neurons. Indeed, $Nek7$ expression was detected in various brain regions[32]. Future work will address how dendritic defects in $Nek7^{gt/gt}$ mice affect the function of the brain and other regions of the nervous system.

## Methods

**Animals and cell cultures**. Pregnant RjOrl:SWISS female mice were purchased from Janvier Labs. For the in vivo analysis of $Nek7$ function $Nek7$ knockout mice (Nek7$^{gt/gt}$) were obtained from EUCOMM as a gene trapped knockout first allele (Nek7$^{tm1a(EUCOMM)Hmgu}$) (http://www.informatics.jax.org/allele/MGI:4888909). All animals were maintained at the Barcelona Science Park (PCB) animal facilities, in strict accordance with the Spanish and European Union regulations. In accordance with applicable legislation, protocols were approved by the Animal Care and Use Committee of the PCB (IACUC; CEEA-PCB).

Hippocampal and cortical cell cultures were prepared from e17.5–18.5 mouse embryos. Hippocampi were dissected, treated with 0.25% trypsin (Life Technologies) during 15 min at 37 °C and dissociated into single cells by gentle trituration. Neurons were seeded on glass coverslips or plastic plates coated with 0.1 mg/ml Poly-D-lysine (Sigma) at ~$10^5$ cells/cm$^2$ or $2 \times 10^4$ cells/cm$^2$ for low-density cultures. Low-density cultures were used to avoid interfering signals from neighboring cells in experiments that required tracing and measurement of axon lengths in 14 DIV neurons or quantification of subcellular distributions of endogenous or plasmid-expressed proteins. Neurons were plated in Dulbecco's Modified Eagle's medium (DMEM) containing 10% fetal bovine serum (FBS), penicillin/streptomycin (pen/strep). After 1–2 h, medium was replaced by Neurobasal medium supplemented with 2% B27, pen/strep, 0.6% glucose, and glutamax (all reagents from Life Technologies). At 3 DIV, cytosine arabinoside (1 µM; Sigma) was added to cultures. One-third of the medium was replaced every 4–5 days. HEK293T, Neuro2A cell lines (both from ATCC; not tested for mycoplasma, not in ICLAC database of commonly misidentified cell lines), or mouse embryonic fibroblasts were grown in DMEM containing 10% FBS. All cells were kept at 37 °C in a humidified atmosphere containing 5% CO$_2$.

**Microarray and qRT-PCR**. For two independent microarray replicas, total RNA was extracted from mouse embryo hippocampal neuron cultures at different time points of differentiation (0 DIV, 20 h, 3 DIV, 6 DIV, 12 DIV, 15 DIV) using a commercial RNeasy minikit (Qiagen) according to the manufacturer's instructions. Then, 25 ng total RNA was amplified using the TransPlex® Complete Whole Transcriptome Amplification Kit (Sigma; reference WTA2) and subsequently labeled using GeneChip Mapping 10K Xba Assay Kit (Affymetrix; catalog no.

900441), according to the manufacturer's instructions. Next, 8 µg of complementary DNA (cDNA) was hybridized per microarray at Mouse Gene 1.0 ST for 16 h of hybridization at 45 °C, and washing and staining of microarrays was performed using a Fluidics Station 450 (Affymetrix, Santa Clara, CA). GeneChips were scanned in a GeneChip Scanner 3000 (Affymetrix, Santa Clara, CA).

For qRT-PCR total RNA was extracted from 0 DIV or 15 DIV cultured hippocampal neurons using RNeasy Mini Kit (QIAGEN), and cDNA was synthesized and amplified from 25 ng total RNA using WTA2 kit (Sigma-Aldrich). Real-time PCR was performed in a CFX96 system (BIO-RAD) from 10 ng double-stranded DNA per reaction using SsoFast Evagreen (BIORAD) following the manufacturer's recommendations. β-Actin was used as house-keeping gene. Normalized expression was quantified using CFXManager software with 2-ΔΔCT method where β-actin is the reference target.

**Plasmids**. The shRNA-expressing plasmids used in this study are based on the pLKO.1 backbone and are part of the Sigma-Aldrich MISSION shRNA collection. The following target sequences were used: shNek7#1: GCATCATTCATTGAGGA TAAT; shNek7#2: CGTGGACAATTTAGTGAAGTT; shNek6#1: CACCGACC TGACATTGTATAT; shNek6#2: CCATCCGAATATCATCAAGTA; shNek9#1: GCTCTGATATCTGTACCTCAT; shNek9#2: CGACAACATCATTGCCTACTA. A pLKO.1 plasmid expressing a scrambled sequence (CAACAAGATGAAGAGC ACCAA) was used as control. A commercial Eg5 shRNA cloned in plasmid pGIPZ was obtained from Dharmacon (RHS4480).

To introduce multiple silent mutations for rendering $Nek7$ shRNA resistant for rescue experiments, or to introduce point mutations to generate the kinase-dead NEK7 (D179A) and the constitutively active (Y97A) mutants of NEK7, PCR mutagenesis was performed on the cDNA of mouse $Nek7$. The wild-type and mutant forms of $Nek7$ were cloned into pCS2 (with N-terminal triple FLAG tag and modified cloning site) using $Fse$I/$Asc$I sites. The non-shRNA-resistant kinase-dead mutant of $Nek7$ (D179A) was cloned into plasmid pCMV5 containing a single FLAG tag.

A cDNA of human KIF11/Eg5[59] was cloned into the modified pCS2-3xFLAG plasmid using restriction enzymes $Fse$I/$Asc$I, and into plasmid mEOS3.2-N1 (gift from Michael Davidson; Addgene plasmid #54525) using $Xho$I/$Eco$RI. The Eg5 S1033A and S1033D mutants were obtained by performing PCR mutagenesis using the wild-type Eg5 plasmid as template. The FLAG-Eg5 cDNAs were cloned into an empty pLL3.7 plasmid using $Nhe$I/$Eco$RI. Plasmid mEos3.2-Tubulin-C-18 was provided by Michael Davidson (Addgene plasmid #57484). The plasmid expressing the CM1 containing fragment of CDK5RAP2 has been described before[15]. The reporter plasmid EB3-tomato was a generous gift of Anne Straube (University of Warwick, UK).

The oligos for genotyping of $Nek7^{gt/gt}$ mice were: 5'-CTGAAGCAGGCCCTG AG (sense) and TCCATTAGCTCACAGTCATTACA (antisense). The primers anneal in positions flanking a loxP site downstream of exon 7 that is only present in the gene trap allele. This results in PCR products that differ in size, depending on the absence/presence of the loxP insertion.

The names and sequences of all the primers used in this study are listed in Supplementary Table 1.

**Antibodies**. The anti-NEK7 antibody was used (3920-1, Epitomics, dilution WB 1:10,000). Other rabbit antibodies used in this study were: anti-Eg5 (GTX30692, GeneTex, dilution IF: 1:250), anti-NEK6 (Ab133494, Abcam, dilution WB: 1:1000), anti-NEK9 (Ab138488, Abcam, dilution WB: 1:1000), and anti-MAP2 (AB5622, Millipore, dilution IF: 1:500). The mouse antibodies used were: anti-α-tubulin (DM1A, Sigma, dilution IF: 1:2000, dilution WB: 1:10,000), anti-β-tubulin (T4026, Sigma, dilution WB: 1:1000), anti-γ-tubulin (TU-30, ExBio, dilution IF: 1:500) anti-acetylated-α-tubulin (6-11B-1, Sigma, dilution IF: 1:50,000), anti-Eg5 (611187, BD Biosciences, dilution WB: 1:1000), anti-Tau (A0024, Dako, dilution WB: 1:10,000), anti-FLAG (F3165, Sigma, dilution IF: 1:2000, dilution WB: 1:10,000), and anti-Histone (gift of Ferran Azorín, IRB Barcelona. Dilution WB:

1:1000). Chicken anti-GFP was used for detection of GFP (Aves Labs, dilution IF: 1:1000).

**Lentivirus production and transduction**. Lentivirus was generated using the LentiLox3.7 system[64]. HEK293T cells were co-transfected with pLKO.1 or pLL3.7 and the packaging plasmids using calcium phosphate. Lentivirus particles in the medium were concentrated 72 h later by ultracentrifugation at 27,000 r.p.m. for 2 h. Virus particles were resuspended in cold phosphate-buffered saline (PBS), aliquoted, and stored at −80 °C. To assay infectivity, HEK293T cells were treated with serial dilutions of concentrated lentivirus, and 72 h after infection, sorted for GFP-positive cells. Neurons were infected at multiplicity of infection (MOI) of 3, and in the case of low-density cultures we used MOI 6.

**Cell culture treatments**. Hippocampal neurons were transfected at the stages indicated in the text and figure legends using Lipofectamine 2000 (ThermoFisher) according to the manufacturer's instructions. In some experiments, as indicated, shRNA plasmids were co-transfected with a EGFP plasmid (pEGFP-N1) in a 5:1 ratio in order to visualize neuron morphology in transfected cells. To perform replating experiments, shRNA-transduced cultures at 5 DIV were trypsinized (0.05% Trypsin-EDTA, ThermoFisher) for 5 min, collected in Neurobasal medium with 5% FBS, pelleted at 800 rpm for 3 min, resuspended in conditioned media, and replated in poly-D-lysine-coated coverslips. After 48 h of regrowth, neurons were fixed and stained for microscope analysis. In Eg5 inhibition experiments, hippocampal neuron cultures were treated at either 3 DIV or 7 DIV with 100 µM monastrol (Sigma), 10 µM STLC (Sigma), 5 µM FCPT (a generous gift from Tim Mitchison, Harvard University, USA), or dimethyl sulfoxide (DMSO) for 6 or 7 days and either live-imaged for EB3-labeled comets or fixed for analysis of dendritic morphology/microtubule acetylation. For EOS live-imaging experiments, neurons were transfected at 3 DIV and analyzed at 9 DIV, and STLC was added 24 h prior to imaging. The mouse neuroblastoma Neuro2A cells were transfected with Lipofectamine 2000 and 72 h later lysed for collection of protein extracts.

**Cell extracts and western blot**. For western blotting, cell culture lysates or hippocampal tissue homogenates were prepared in Lysis buffer (50 mM Tris pH 7.4, 150 mM NaCl, 1 mM MgCl₂, 1 mM EGTA, 10% Glycerol, and 1% Triton X-100) in the presence of protease inhibitors (Complete, Roche). Lysates were clarified by centrifugation at $16,000 \times g$ for 10 min at 4 °C. To obtain soma or axonal protein extracts, hippocampal neurons were cultured on filters with 3 µm pore size membranes (Neurite Outgrowth Assay Kit, Millipore). At 8 DIV, cells were fixed in methanol at −20 °C for 5 min. To obtain axonal fractions, somas and dendrites on the upper side of the device were first carefully removed with wet flattened cotton swabs following the manufacturer's instructions. Axons at the bottom side of the membrane were extracted in a 200 µl drop of sodium dodecyl sulfate (SDS) sample buffer. To obtain somato-dendritic fractions, axons from the bottom part of the device were first removed by scraping. Then, SDS sample buffer extracts were prepared from the upper side of the membrane. All samples were clarified by centrifugation and boiled in sample buffer for SDS–polyacrylamide gel electrophoresis. Proteins were separated and transferred onto polyvinylidene difluoride membranes (Millipore). Membranes were blocked with 5% milk powder in TBST (20 mM Tris, pH 7.5, 150 mM NaCl, 0.05% Tween-20) for 1 h and probed overnight with primary antibodies diluted in tris-buffered saline with Tween-20 (TBST). The relative intensity of the bands was quantified using Quantity One 1-D analysis software (Bio-Rad Laboratories). Uncropped images of western blots are shown in Supplementary Figs. 7–9.

**Immunofluorescence microscopy**. Cultured neurons and cell lines were fixed in methanol at −20 °C for 5 min or in 4% paraformaldehyde (PFA)/4% sucrose diluted in PBS for 15 min at room temperature. For staining Eg5, cells were simultaneously permeabilized and fixed using 4% PFA/4% sucrose/0.25% glutaraldehyde/0.1% Triton X-100 diluted in PHEM buffer (60 mM Pipes, 25 mM Hepes pH 7.4, 5 mM EGTA, 1 mM MgCl₂). All fixed cells were permeabilized with 0.25% Triton X-100 in PBS for 5 min, blocked with 4% bovine serum albumin (BSA, Sigma) and incubated in 2% BSA overnight with primary antibodies. Alexa350, Alexa488, or Alexa568 secondary fluorescent antibodies were used (ThermoFisher). Nuclei were stained with DAPI (4′,6-diamidino-2-phenylindole). Different samples within one experiment were imaged using constant intensity and exposure settings, avoiding signal saturation. A confocal microscope TCS-SP5 (Leica Microsystems) equipped with a 63×/1.40 OIL objective was used for imaging dendritic spines at 1024 × 1024 pixel resolution. Dendritic spine image stacks were taken with a 0.125 µm step-size, and deconvolved using ImageJ software (NIH). For analysis of Eg5 stainings, single-plane images of neuron somas and dendrites were acquired with an Orca AG camera (Hamamatsu) coupled to a Leica DMI6000B microscope equipped with a 100×/1.40 OIL objective. 20×/0.50 DRY and 40×/1.25 OIL objectives were additionally used for standard imaging and for mosaics generation of complete axons/dendritic arbors and replated neurons. "Proximal dendrite region" corresponds to a distance of ~50 µm from the soma, while "distal dendrite region" corresponds to a distance of ~50 µm from the tip.

**Time-lapse microscopy**. Hippocampal cultures were plated in glass-bottom dishes (MatTek), transduced with virus expressing shRNA, or treated with drugs, transfected with EB3-tomato reporter at 7 or 8 DIV and imaged 24 h later, or with mEOS3.2 constructs at 5 DIV and imaged 3 days later. For analysis after expression of CDK5RAP2 fragment 51–100, cultures were co-transfected with plasmid encoding EGFP, or EGFP-CDK5RAP2 fragment 51–100 together with EB3-tomato reporter (2:1 ratio) at 6 DIV and imaged 48 h later. Live imaging of EB3 comets or mEOS3.2 alone or fused to Eg5 or α-tubulin was performed in the dendrites of random transfected cells, using an Olympus IX81 microscope equipped with Yokogawa CSU-X1 spinning disc and a temperature-controlled CO₂ incubation chamber. Image stacks were acquired with 100×/1.4 OIL immersion objective and an iXon EMCCD Andor DU-897 camera, using iQ2 software. Fluorescent images with pixel size of 0.14 µm were taken at 1 s intervals during 2.5 min for EB3 comets in dendrites. Fluorescent images with pixel size of 0.14 µm were taken at 10 s intervals during 4 min for mEOS experiments. mEOS3.2 was photoconverted using a 405 nm laser at 5% for 1 s.

**Image analysis**. All images were processed and quantified using the ImageJ software (NIH). Blinding was used in the sample imaging and image quantifications. For all fluorescence intensity measurements, background signal was measured in an adjacent area and subtracted. Dendritic EB3 comet analysis was performed using the kymograph macro (ImageJ software), with lines drawn on the trajectories of comets. The spatial distribution of mEOS3.2 alone or of tagged proteins was measured using the Dynamic ROI Profiler plugin (ImageJ software). The average fluorescence intensities of α-tubulin and acetylated α-tubulin staining were measured along the imaged dendritic trace. Whole axon and dendrite length were measured using the NeuronJ macro (ImageJ software). Sholl analysis was performed using the Sholl analysis plugin[65] using binary versions of the dendrite tracings generated with the NeuronJ macro. Dendritic spine morphology and density analysis was performed along the principal dendrite (thickest dendrite at 14 DIV) before the first branching, on deconvolved image stacks using NeuronStudio Software[66] or IMARIS software (BITPLANE).

**Tissue preparation and biolistics**. Adult (3/4 months old) animals were killed through intracardiac perfusion. A quick perfusion was applied, consisting of an initial 10 ml of PBS 0.1 M, followed by 40 ml of PFA 4% (20 ml/min). The brains were extracted from the perfused animals, incubated for 10 min in PFA 4% at 4 °C, and then stored in PBS at 4 °C for up to 48 h. The brain tissue was embedded in agarose (4% in PBS) and cut in 100 µm coronal slices using a vibratome. Slices were stored at −20 °C in cryoprotective medium, and washed thoroughly with PBS before shooting with the gene gun. DiI-tungsten-coated bullets were shot on the tissue sections by using a Helios Gene Gun (Bio-Rad #165–2431) for neuron labeling, working at 120 psi of helium gas pressure, and using a membrane filter between the gun and tissue[67]. After delivery of particles, the tissue sections were incubated and thoroughly washed in PBS and then stained with Hoechst 1:500 for 1 h at room temperature, once again washed in PBS, and mounted on slides with Mowiol mounting medium (Sigma).

**Statistical analysis**. Statistical analysis was done using the Prism 6 software. Two-tailed unpaired t-tests were performed to compare the experimental groups. The results are detailed in the figures and the figure legends. Sample sizes were determined based on experience from previous work and based on other similar published studies. The number of Nek7ᵍᵗ/ᵍᵗ embryos and adult animals used in this study was as high as possible (due to sub-Mendelian birth rates). No data were excluded and samples/cells were randomly allocated to experimental groups.

Microarray raw expression files (CEL format) from all samples were normalized using the "rma" function from the *oligo* R package[68,69] summarizing to probeset level. For each transcript, the 50% most variable probes (interquantile value) were selected. Gene expression was summarized by applying median polish to the selected probesets (function "medpolish" of the R *stats* package).

Technical effects due to experimental batches were adjusted by fitting a linear model ("lmFit" of the *limma* package[70]) for each gene with time and batch as covariates and removing the estimated coefficient for date of experiment. Samples corresponding to times 20 h and 2 days were combined through the mean expression of each gene. The resulting matrix was centered and scaled.

Clusters of genes with similar behavior along time points were found using the function "cutHclust" of the R *ClassDiscovery* package from the OOMPA suite (http://oompa.r-forge.r-project.org) with the "complete" linkage method and "pearson" metric. The number of clusters was set to 20.

**Data availability**. The authors declare that all the data supporting the findings of this study are available within the article and its supplementary information files. All primary microarray data are available from the Gene Expression Omnibus (GEO) database: GSE113680.

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

## Acknowledgements

We are grateful to Anne Straube (University of Warwick, UK), Ferran Azorin (IRB Barcelona, Spain), Carme Gallego (IBMB, Barcelona, Spain), Michael Davidson, and Tim Mitchison (Harvard University, USA) for providing reagents. This study was supported by grants BFU2009-08522, BFU2012-33960, and BFU2015-69275-P (MINECO/FEDER) to J.L., BFU2014-58422-P to J.R., and SAF2016-76340-R to E.S. (MINECO, Spain), and IRB Barcelona intramural funds. J.L. and Y.M. acknowledge additional support from the Ramón y Cajal and Juan de la Cierva Programmes, respectively (MINECO, Spain). F.F. was supported by a La Caixa PhD fellowship ('La Caixa' Foundation, Spain).

## Author contributions

F.F. designed and performed most experiments, analyzed data, and wrote the paper. P.M.D. contributed to all experiments involving $Nek7^{gt/gt}$ animals, performing breedings, genotyping, western blotting, and helped in the preparation of animals for experiments. Y.M. and F.F. perfused animals, prepared bullets, and performed gene gun shootings. C.S.-H. assisted F.F. in planning and execution of functional studies using neuronal cultures. C.L. prepared neuronal cultures and extracted RNA for microarray analysis. J.R. provided various plasmids and expertise throughout this study related to the functions of NEK6, NEK7, NEK9, and Eg5, and the $Nek7^{gt/gt}$ animals. E.S. provided reagents and expertise for the analysis of dendrites in vivo. J.L. designed the study, analyzed data, and wrote the paper.

## Additional information

**Competing interests:** The authors declare no competing interests.

