## [Peer Review File · Nature Communications]

Reviewers' comments:

Reviewer #1 (Remarks to the Author):

In this manuscript, Freixo and colleagues attempt to elucidate how microtubules are ordered in differentiated cells and how this order contributes to cellular architecture, using neurons as model system. Analysis of (published/public) microtubule proteome data from cycling cells led to the identification, after genome microarray of cultured hippocampal neurons, of potential microtubule regulators in neurons. For the final selection, authors performed a mRNAi screen, looking for morphological phenotypes. One of the genes that produced strong phenotype in differentiated neurons was the mitotic kinase NEK7. The knockdown experiments (Fig. 1) convincingly show that NEK7 and not the related kinases NEK6 and NEK9 plays a role in the late, but not early, process of dendritic growth. Moreover, the catalytically inactive form of NEK7 failed to rescue the shRNA phenotype, indicative that the kinase activity is required for morphological differentiation. Authors then go on and demonstrate that NEK7 works via the kinesin Eg5/Kif11: i) knockdown on this kinesin phenocopies the NEK7 knockdown phenotype and ii) the phosphomimetic Eg5 rescues NEK7 loss. Several other evidences are presented reinforcing the view that NEK7-Kif11 plays a key role in dendritic growth via microtubule distal organization, including in vivo data. This is an excellent study, suitable for publication at Nature Communications. It would be important if authors could address the following comments.

Introduction. Please shorten this long paragraph (line 59 to 86) on microtubule organisation in cycling cells etc. This would be fine for a review or commentary but here it is a bit redundant (in this long form). I suggest to make this shorter and add a paragraph on microtubule and dendritic branching which is the actual focus of the study.

Results.

Fig. 1 d and Fig. 1 e. These data are not very relevant and can be moved to supplementary.

Fig. 2 e and Fig. 2 g. The spine analysis is interesting but not all that relevant for the study, somehow perturbing the flow of the study. I would recommend removing these data or just making a brief mention and move to supplementary.

Fig. 3. It would be of interest to know if, like NEK7 (Fig. 1f), Kif11 is polarised to the dendritic fraction of the cultured neurons.

Fig.4. It is really hard to see the distribution of Kif11 in control and in NEK7 depleted neurones. The staining is weak and looks rather diffuse and not particularly specific. It is also unclear how the reduction of NEK7 would lead to the distal reduction of this kinesin without a concomitant increase in proximal regions (assuming that kinase activity of NEK7 is required for the anterograde movement of the motor). All in all, the data in this figure are not sound, not reflecting the described results.

Reviewer #2 (Remarks to the Author):

In this work, Freixo and colleagues find that the mitotic kinase NEK7 can regulate the morphogenesis of dendrite by phosphorylating Eg5/KIF11 which can promote the stabilization of MTs. By performing a genome microarray analyze in different stages cultured hippocampal neurons, the authors find that NEK7 expression is upregulated during development and NEK7 knockdown cells show both dendrite branching and dendritic spine. Consistent with the findings in mitosis, NEK7 can also phosphorylate Eg5/KIF11 in neurons and promote its localization in distal dendrite. Eg5/KIF11 might be able to stabilize MT by its MT binding activity but not motor activity thus to maintain the MT polarity in the distal dendrite. They propose that the altered MT polarity in Eg5 or NEK7 knockdown cell might lead to a reduced dendrite length. However, the in vivo knockout phenotype appears to be quite subtle. I feel that more experiments should be done with knockout cells and in knockout animals to make sure that the model is correct.

Main points:

1) For the reduced localization of Eg5 in distal region when treat cell with shNek7 in Fig. 4, the author should check if the total expression of Eg5 changed? In other words, does cell body level of Eg5 change in Nek7 knockdown cells? The authors should also present data showing the localization of Eg5 in axon and whether it is affected by Nek7?

It should also be noted that drawing conclusions of protein subcellular distribution using overexpressed transgenes in a cultured system is some time difficult because of the potential overexpression artifacts. Since this is such an important part of this paper, I suggest that these experiments in Fig. 4 be done using the nek-7 knockout animals and it will be better to do it in vivo.

2) In Fig. 3, the expression level of the Eg5 S1033D construct is at least several fold lower than the wt. could the strong rescue effect of this variants due to expression at a different level?

3) For Fig. 5 and Fig. 6, could the author explain why the STLC treatment specifically affects the MT polarity or stabilization in the distal region while Eg5 is more concentrated in proximal region? This is an important prediction of the model. Does the treatment affect the localization of Eg5? Could FCPT treatment rescue the defect of shNEK7 treated cell? And how could the author exclude the non-specific effects of C5R2 51-100 overexpression?

4) As the NEK7 knock out mouse shows similar dendrite defect with the culture cell, the author should check the MT polarity and Eg5 localization in the cell prepared from knock out mice.

Will the MT polarity change be earlier than dendrite defect occurrence in knock out mouse?

5) The dendrite branch distribution of WT in Fig. 8c looks different to the WT in Fig. 1k. Is there no defect in the distal region of Nek7 mutant animals?

Minor points:

1) For Fig. 3c, the complexity of dendrite should also be presented.

2) In Fig. 2e, the difference between rescue and shNek7 is more obvious than the difference between control and shNek7, but the significance is opposite. The similar situation as Fig. 2f.

We would like to thank the reviewers for their detailed analysis of our manuscript. They have provided very helpful comments that we have addressed with new experiments and with various text changes. We believe that this has led to significant improvements in our revised manuscript. Below is our point-by-point response explaining in detail how we have addressed the specific issues raised by the reviewers.

Reviewer 1:

In this manuscript, Freixo and colleagues attempt to elucidate how microtubules are ordered in differentiated cells and how this order contributes to cellular architecture, using neurons as model system. Analysis of (published/public) microtubule proteome data from cycling cells led to the identification, after genome microarray of cultured hippocampal neurons, of potential microtubule regulators in neurons. For the final selection, authors performed a mRNAi screen, looking for morphological phenotypes. One of the genes that produced strong phenotype in differentiated neurons was the mitotic kinase NEK7. The knockdown experiments (Fig. 1) convincingly show that NEK7 and not the related kinases NEK6 and NEK9 plays a role in the late, but not early, process of dendritic growth. Moreover, the catalytically inactive form of NEK7 failed to rescue the shRNA phenotype, indicative that the kinase activity is required for morphological differentiation. Authors then go on and demonstrate that NEK7 works via the kinesin Eg5/Kif11: i) knockdown on this kinesin phenocopies the NEK7 knockdown phenotype and ii) the phosphomimetic Eg5 rescues NEK7 loss. Several other evidences are presented reinforcing the view that NEK7-Kif11 plays a key role in dendritic growth via microtubule distal organization, including in vivo data. This is an excellent study, suitable for publication at Nature Communications. It would be important if authors could address the following comments.

Response: We thank the reviewer for this well-balanced summary and overall very positive evaluation of our work. We have addressed the reviewer's comments below:

Introduction. Please shorten this long paragraph (line 59 to 86) on microtubule organisation in cycling cells etc. This would be fine for a review or commentary but here it is a bit redundant (in this long form). I suggest to make this shorter and add a paragraph on microtubule and dendritic branching which is the actual focus of the study.

Response: We agree and have followed the reviewer's suggestion: we have significantly shortened the part describing centrosome duplication and mitotic microtubule organization and have added a brief description of the role of microtubules in dendrite morphogenesis.

Results.

Fig. 1 d and Fig. 1 e. These data are not very relevant and can be moved to supplementary.

Response: In our opinion the data is relevant since it demonstrates that NEK7 protein levels are upregulated in vivo, a result that nicely complements our very similar observation made in neurons cultured in vitro (Fig. 1b, c). At this point there

are no space constraints in Fig. 1, which is why we would rather keep this data in the main figure.

Fig. 2 e and Fig. 2 g. The spine analysis is interesting but not all that relevant for the study, somehow perturbing the flow of the study. I would recommend removing these data or just making a brief mention and move to supplementary.

Response: Indeed, the spine phenotype is not the focus of our study. However, its analysis is an important part of the characterization of NEK7's role in neurons, which has never been described before. In Fig. 8 we show that the spine defects are also observed in vivo. Together the spindle defects may have implications for further studies, because, as we show, NEK7 seems to regulate spine morphology independently of Eg5 (Fig. 2d, f; Suppl. Fig. 3e, g) and thus other relevant substrates are yet to be identified.

Fig. 3. It would be of interest to know if, like NEK7 (Fig. 1f), Kif11 is polarised to the dendritic fraction of the cultured neurons.

Response: This is an interesting point. We have tested this by probing lysates prepared from axonal and somato-dendritic fractions as in Fig. 1f with Eg5 antibody by western blotting. Indeed, we found that Eg5 is enriched in the somato-dendritic fraction, similar to what we observed for NEK7. We have included this data in Suppl. Fig. 3h.

Fig.4. It is really hard to see the distribution of Kif11 in control and in NEK7 depleted neurones. The staining is weak and looks rather diffuse and not particularly specific. It is also unclear how the reduction of NEK7 would lead to the distal reduction of this kinesin without a concomitant increase in proximal regions (assuming that kinase activity of NEK7 is required for the anterograde movement of the motor). All in all, the data in this figure are not sound, not reflecting the described results.

Response: Indeed, Eg5 staining appears diffuse and the changes in its distribution in neurons are inherently difficult to appreciate by eye. However, our quantification clearly reveals reduction of the Eg5 signal after NEK7 depletion and restoration of the signal in cells expressing NEK7 wild type or Eg5S1033D phospho-mimetic mutant. For the antibody used in these stainings we previously showed that it labelled, as expected, the centrosome and spindle in cycling cells. We provide direct evidence for the specificity of the Eg5 staining in neurons: expression of Eg5 shRNA, which we found by western blot to deplete Eg5 in mouse cells, leads to a reduction in the dendritic Eg5 staining produced by this antibody (Suppl. Fig. 4d-g). Altogether the data demonstrates the specificity of the immunofluorescence signal produced by the Eg5 antibody in neurons. In addition, to improve the visual assessment of changes in Eg5 distribution, we have improved contrast and presentation of the immunofluorescence images showing Eg5 staining (Fig. 4; Suppl. Fig. 4; Suppl. Fig. 5; Suppl. Fig. 6).

Regarding the Eg5 levels in different dendrite regions: our data suggests that Nek7 phosphorylation impairs Eg5 localization throughout the dendrites, most strongly in, but not restricted to, distal regions (as revealed by our quantifications in Fig. 4). Thus, if one were to expect NEK7 depletion to cause an increase of Eg5 in a certain compartment concomitant with its decrease in dendrites, it may be in the

soma. Here we detected only a slight, non-significant increase of Eg5 signal (Fig. 4c). However, in similar quantifications performed in Nek7 k.o. neurons (new experiments presented in Suppl. Fig. 6h, i, j; see also reviewer #2, point 1 below), which are expected to completely lack NEK7 activity, we do see a significantly increased signal of Eg5 in the soma concomitant with its reduction in the dendrites. We conclude that efficient reduction of NEK7 activity strongly impairs Eg5 accumulation in dendrites, and the resulting redistribution of Eg5 leads to slightly increased Eg5 levels in the soma.

Reviewer #2:

In this work, Freixo and colleagues find that the mitotic kinase NEK7 can regulate the morphogenesis of dendrite by phosphorylating Eg5/KIF11 which can promote the stabilization of MTs. By performing a genome microarray analyze in different stages cultured hippocampal neurons, the authors find that NEK7 expression is upregulated during development and NEK7 knockdown cells show both dendrite branching and dendritic spine. Consistent with the findings in mitosis, NEK7 can also phosphorylate Eg5/KIF11 in neurons and promote its localization in distal dendrite. Eg5/KIF11 might be able to stabilize MT by its MT binding activity but not motor activity thus to maintain the MT polarity in the distal dendrite. They propose that the altered MT polarity in Eg5 or NEK7 knockdown cell might lead to a reduced dendrite length. However, the in vivo knockout phenotype appears to be quite subtle. I feel that more experiments should be done with knockout cells and in knockout animals to make sure that the model is correct.

Response: To address this reviewer's concern and provide more evidence for the proposed role of NEK7 in dendrite and spine morphogenesis, we have performed several new experiments, which we will describe in our point-by-point response below.

Main points:

1) *For the reduced localization of Eg5 in distal region when treat cell with shNek7 in Fig. 4, the author should check if the total expression of Eg5 changed? In other words, does cell body level of Eg5 change in Nek7 knockdown cells? The authors should also present data showing the localization of Eg5 in axon and whether it is affected by Nek7?*

Response: We have addressed all of these points. We have added a western blot analysis showing that the total levels of Eg5 are similar in control and NEK7-depleted neurons (Suppl. Fig. 4c). Moreover, as described above in response to reviewer #1, we observed that the reduction of Eg5 in dendrites indeed results in a slight increase of Eg5 signal in the cell body. This redistribution of the Eg5 signal is most evident in Nek7 k.o. neurons (new experiment presented in Suppl. Fig. 6h, i, j) but less so after treatment with Nek7 shRNA, presumably because the knock-down approach is less efficient and causes only partial redistribution of Eg5. As requested, we have also tested whether NEK7 affects Eg5 localization in axons. Our quantifications show that the axonal Eg5 signal is not affected by NEK7 knock down (Suppl. Fig. 4h, i).

Altogether the results support our conclusion that NEK7 controls Eg5 localization specifically to the dendritic compartment, in particular in distal regions.

It should also be noted that drawing conclusions of protein subcellular distribution using overexpressed transgenes in a cultured system is some time difficult because of the potential overexpression artifacts. Since this is such an important part of this paper, I suggest that these experiments in Fig. 4 be done using the nek-7 knockout animals and it will be better to do it in vivo.

We agree with the reviewer that overexpression may alter subcellular distribution of proteins. For this reason, for the analysis of the distribution of Eg5 after NEK7 knockdown, the most important experiment in Fig. 4, we quantified endogenous Eg5, not overexpressed Eg5 (Fig. 4a-d). Then, in Fig. 4e and 4f, we analysed the distribution of wild type Eg5 vs. phosphorylation mutants, which required exogenous expression. Here we would like to emphasize that NEK7 depletion reduces the intensity of the Eg5 signal in dendrites for both endogenous and exogenous Eg5, suggesting that the expression levels of exogenous Eg5 do not interfere with this regulation. We believe that this is due to the relative moderate expression of recombinant Eg5 in these cells. To demonstrate this we cloned the Eg5 cDNAs into lentiviral vectors to allow infections of neuronal cultures for subsequent western blot analysis (new Fig. 3a). This method routinely achieves expression of the exogenous constructs in close to 100% of the cells. This experiment provided two important pieces of information: first, none of the recombinant Eg5 proteins is expressed strongly above the levels of endogenous Eg5 (see Eg5 western in Fig. 3a), indicating that there is indeed no significant overexpression of these constructs in neurons. Second, wild type and mutant Eg5 proteins are all expressed at similar levels (see FLAG western in Fig. 3a), suggesting that the observed differences in their ability to rescue NEK7 depletion (Fig. 4e, f) are not due to differences in their relative expression levels.

Regarding the reviewer's suggestion to use knockout animals we would like the reviewer to consider the following points: i) while the use of knockout animals eliminates variability introduced by transfection of shRNA, it does not address the reviewer's concern regarding overexpression of Eg5 variants; ii) as we describe in our manuscript (p. 18/19) and consistent with previous reports (Shi et al. 2016, Nat Immunol; Salem et al. 2010, Oncogene), NEK7^{-/-} animals are born at sub-Mendelian ratio and thus only very few homozygous *Nek7* knockout animals can be obtained for experiments. iii) quantification of the Eg5 distribution in dendrites in vivo using immunohistochemistry in tissue sections would be technically much more challenging than in cultured cells and, even if feasible, accuracy would most likely be lower.

Based on these considerations we decided to use the few *Nek7* knock out animals that we could obtain during the revision period for deriving neurons and performing new experiments on these cells in culture. We found, in agreement with the knock-down experiments, Eg5 staining in dendrites of *Nek7* k.o. neurons was reduced, in particular in distal regions, when compared to neurons prepared from wild type controls. In addition, as mentioned above in response to reviewer #1, *Nek7* k.o. neurons displayed a slight increase in the Eg5 signal in the soma compared to controls, consistent with a re-distribution of Eg5 from the dendrites to the cell body. This new data was included in Suppl. Fig. 6h, i, j.

2) *In Fig. 3, the expression level of the Eg5 S1033D construct is at least several fold*

lower than the wt. could the strong rescue effect of this variants due to expression at a different level?

Response: The western blot in Fig. 3 was performed in Neuro2A cells to show that all constructs have similar expression levels. It seems that the reviewer refers to the anti-Eg5 western, which shows some differences between samples. However, the anti-FLAG western (which detects specifically the exogenous Eg5 proteins) shows very similar levels. To clarify this issue we have expressed all Eg5 variants in neurons by viral infection and re-analyzed their expression levels by western blotting. As explained in our response to point 1 above, the side-by-side comparison of the samples did not reveal any significant differences in expression levels. We have replaced the previous, potentially misleading western blot with this new, more relevant and accurate analysis (new Fig. 3a).

3) For Fig. 5 and Fig. 6, could the author explain why the STLC treatment specifically affects the MT polarity or stabilization in the distal region while Eg5 is more concentrated in proximal region? This is an important prediction of the model. Does the treatment affect the localization of Eg5?

Response: Regarding the distribution of Eg5 in untreated neurons, our data suggest a relatively even distribution along the dendrite. In wide-field microscopy the greater thickness of proximal compared to distal dendrites, may give the impression of an increased Eg5 signal, however, in confocal images the intensities of Eg5 in proximal and distal regions are similar (see quantification below).

While the STLC treatment affects microtubules in distal dendrite regions more strongly, both in terms of polarity and stability, we would like to emphasize that the effect is not specific to the distal region, as phrased by the reviewer, since we also observe similar effects in the proximal dendrite region, albeit less pronounced. In fact this is very consistent with the results from neurons depleted of NEK7.

To answer the reviewer's question regarding the effect of STLC on Eg5 localization, we treated neurons with STLC and quantified Eg5 intensities. We found that STLC treatment reduced Eg5 staining throughout dendrites both in proximal and distal regions when compared to control neurons, consistent with a general weakening of the interaction of Eg5 with microtubules (Suppl. Fig. 5d, e, f). This reduced binding to microtubules was also observed in our photoconversion experiments, in which addition of STLC eliminated the immobile fraction of Eg5-EOS (Fig. 7). The question remains, why STLC treatment, similar to depletion of NEK7,

affects microtubule polarity and stability more strongly in distal regions? STLC treatment has a microtubule destabilizing effect due to the displacement of Eg5 from microtubules. Since microtubules in the distal part of dendrites were shown to be overall less stable compared to more proximal microtubules, as judged by staining intensities for acetylated tubulin (stable microtubules) and tyrosinated tubulin (recently assembled microtubules) (Kollins et al., 2009, Neural Dev) they may be more susceptible to the effects of STLC, which may explain the more pronounced effects in distal dendrites. We performed a similar analysis of the distribution of stable (acetylated) microtubules in dendrites as in the references above under our experimental conditions by quantifying intensities of acetylated over total microtubule staining (included as new Suppl. Fig. 4j, k). Indeed we confirmed that in DIV9 neurons the amount of acetylated microtubules relative to total microtubules is reduced in distal compared to proximal dendrite regions.

Could FCPT treatment rescue the defect of shNEK7 treated cell?

Response: This is an interesting possibility and we have tested it. As can be seen in new Suppl. Fig. 6a-e, FCPT treatment significantly increased dendrite length in NEK7-depleted neurons, but the effect was relatively mild and no full rescue was achieved. Regarding the spine defects in NEK7-depleted neurons, FCPT rescued spine density, but had no effect on morphology. These results suggest that microtubule stabilization by Eg5 (enhanced by FCPT) is an important contributor to its function, but the ATP-dependent motor activity also plays an important role. Apart from the description of this new experiment in the result section, we have also included a short discussion of this finding.

And how could the author exclude the non-specific effects of C5R2 51-100 overexpression?

Response: The 51-100 fragment of CDK5RAP2 comprises the CM1, a conserved sequence motif, which in yeast was shown to directly interact with two different γ TuRC subunits, GCP3 and MZT1 (Lin et al. 2016, JCB). The specificity of this interaction has allowed the use of the CM1 fragment for purifying γ TuRC from whole cell extract and for stimulating γ TuRC nucleation activity in vitro and in vivo (Choi et al., 2010, JCB). Considering these findings and the small size of the CM1-containing fragment (~50 aa) it is reasonable to assume that the effects of its overexpression are specific to γ TuRC. While, one can never completely rule out any unspecific effects in this type of experiment, the features of the CM1 fragment minimize this risk.

4) As the NEK7 knock out mouse shows similar dendrite defect with the culture cell, the author should check the MT polarity and Eg5 localization in the cell prepared from knock out mice. Will the MT polarity change be earlier than dendrite defect occurrence in knock out mouse?

Response: This is an important request and we have performed this experiment. We prepared neuronal cultures from *Nek7^{gt/gt}* embryos and quantified both Eg5 localization (as already discussed above) and microtubule polarity. In both analyses of *Nek7* k.o. neurons we could recapitulate our observations made with shRNA-treated neurons (new Fig. 8g, h, i; new Suppl. Fig. 6h, i, j). We also tested whether the change in microtubule polarity preceded the defect in dendrite length. Indeed,

dendrite length was not yet significantly affected at 9DIV (Suppl. Fig.6k, l), the time point of the polarity measurements, but became apparent 5days later at 14DIV (Fig. 8a, b)

5) *The dendrite branch distribution of WT in Fig. 8c looks different to the WT in Fig. 1k. Is there no defect in the distal region of Nek7 mutant animals?*

Response: We have added new quantifications of dendrite branching from additional animals to the graph in Fig. 8c, which makes this data more robust. However, we still observe a difference in the profiles when comparing with the experiments in Fig 1k. However, we would like to point out that these two analyses (Fig. 1 vs. Fig. 8) are not absolutely comparable in the sense that they were done with neurons from two different mouse strain backgrounds, in one case (Fig. 1) with transfection, in the other case (Fig. 8) without transfection, and with approximately 2 years between the two analyses. Thus, while not being directly comparable, the analyses are consistent with respect to the clear reduction in dendritic complexity that we observed for both NEK7-depleted and Nek7 k.o. neurons relative to their respective controls.

Minor points:

1) *For Fig. 3c, the complexity of dendrite should also be presented.*

Response: We have re-analysed the samples and included a Sholl analysis as a new panel (Fig. 3d). As expected, only the phospho-mimetic S1033D mutant rescues the dendritic branching defects caused by NEK7 depletion.

2) *In Fig. 2e, the difference between rescue and shNek7 is more obvious than the difference between control and shNek7, but the significance is opposite. The similar situation as Fig. 2f.*

Response: The reviewer is right, but this is not a contradiction. The mean values for two samples may be very different, but depending on the distribution of the individual data points the statistical significance may not be that high. For the same reason the mean values of two samples may not differ greatly, but the difference can still be highly significant.

Reviewers' comments:

Reviewer #1 (Remarks to the Author):

The authors have responded satisfactorily to my suggestions. I also accept the refusal to follow one of my recommendations. For my part this work can be published in Nature Communications

Reviewer #2 (Remarks to the Author):

The authors have performed a number of new experiments and have improved the manuscript. This is a comprehensive analyses of the relationship between a microtubule regulating enzyme and dendrite development.

I unfortunately still have reservations of this manuscript for the following reasons:

1. The molecular links between NEK7, Eg5 and MT are known. This is not a real problem for this paper since the key questions is whether this pathway is involved in dendrite branching and how it does it.
2. The phenotypes. In vitro, the phenotypes are consistent with each other. However, the shape of these cultured neurons are dramatically different between Fig. 1 and Fig. 8. This does raise the question about how physiological this prep is in terms of assessing dendrite development. The fact that the in vivo development phenotype is extremely subtle Fig. 8i is alarming. I requested to do more in vivo analyses but the authors argued that they couldn't get enough animals.
3. the mechanisms. If we believe that there is a real phenotype here, then we can try to assess if the mechanism is novel and interesting. There are EB1 phenotypes in distal dendrite, but not in proximal dendrites. This difference is not satisfyingly explained. It is also not clear why retrograde MT polymerization should be inhibitory to dendrite growth, as cited in the abstract. So at the end, I felt these data still did not lead to a clear model of MT and dendrite growth.

Reviewer #3 (Remarks to the Author):

This is a thorough study on the roles of Nek7 and Eg5 on dendrite and spine development, where effects of both of these proteins are characterized in cultures and Nek7's role is also studies a knockout mouse model. Authors carefully selected Nek7 to study based on its developmentally regulated expression level. They find that Nek7 and Eg5 are required for appropriate dendrite and spine formation. The manuscript is interesting and relevant for the neuroscience audience. The translation of microtubule regulatory machinery which is initially discovered in dividing cells to postmitotic neurons is very nice. Authors provide evidence showing that Nek7 alters dendritic distribution of Eg5. Nek7's phosphorylation of Eg5 is key

for dendritic distribution. Both Nek7 and Eg5 alter microtubule dynamics in distal dendrites and Eg5 activity regulates microtubule stability. The effect of STLC on Eg5's microtubule binding is supported by EOS-Eg5 phosphoconversion experiments.

Overall culture data provides evidence on Nek7 regulating microtubule dynamics and dendrite development as well as Eg5 localization and separately it provides data on Eg5's role on microtubule dynamics and microtubule stability using drugs targeting Eg5. There might be multiple effectors of Nek7 and it is not surprising that Nek7 shRNA effects are not rescued by opposing effects of FCPT. Together with S1033D mutant's rescue of dendrite deficits caused by Nek shRNA, study supports that Nek's effects are partially mediated by Eg5.

There are sections of the study that I think should be better supported with experiments and/or analysis.

Major comments

1- Authors use STLC and FCPT to alter Eg5 function. How specific are these molecules? Could they target other kinesins? They have generated Eg5 shRNA, which reduces endogenous Eg5 levels (a key reagent in the study). Does the Eg5 shRNA reduce dendrite length similar to STLC? Authors should either test the effect of Eg5 shRNA on dendrite morphogenesis (the most prominent phenotype) or authors should confirm that the effect of pharmacological treatments (STLC and FCPT) are absent in Eg5 shRNA expressing neurons, confirming their specificity.

2- Authors' hypothesis of Nek7 regulating Eg5 on Ser1033 is based on previous studies, which nicely show the phosphorylation being affected by Nek7. The Nek7 KO mice is a very useful reagent to test this finding in vivo. Authors should request the Ser1033 phosphospecific antibody and test if in mouse brain lysates the phosphorylation levels are changed or not. Phosphospecific antibodies often are not specific in immunostaining so I would recommend Westerns for this experiment.

3- Fig 4. The phosphomimetic mutants D or E are usually ineffective as mimics, but as in this example, rarely they act as true phosphomimics. In Figure 4a and b, authors normalized the endogenous Eg5 intensity by dividing dendrite with soma intensity. In figure 4e-f authors normalized overexpressed Eg5 intensity with the GFP intensity. I think for all quantifications the Eg5 measurements should be normalized with GFP (to control for volume changes) and also with the soma intensity (to control for the total levels of Eg5 in that particular cell). This way we can have a measure of the localization of Eg5 to dendrites. For Figure 4a-b authors did not normalize for GFP, however later on in 4d they state that there is no overall difference between different conditions in GFP dendrite/soma ratio. Therefore, it is unlikely that observed changes in 4b would be due to volume change. So, 4a-b is convincing, as it is. For 4e-f authors state that they have normalized with GFP in order to compensate for the cell to cell variability of overexpressed Eg5. However, expression levels of GFP and Eg5 are independent, a cell may have high levels of GFP and low levels of Eg5. Authors should also quantify in 4f dendrite/ soma of GFP normalized overexpressed Eg5 intensities.

4- In Fig 3 and 4, about overexpression of wt and mutant Eg5: 3a shows similar expression levels of mutants and wt and that these levels are similar to endogenous Eg5. This piece of evidence does not mean that when rescue experiments are performed in 3b-d or 4 e-f, the lipofectamine transfected neurons' Eg5 levels were similar. There is large variation in

expression levels of lipofectamine transfected neurons, more variability than lentivirus expression. For Fig 3d, authors should clarify how they have selected neurons to image (did they check FLAG expression levels?) For 4e-f, as mentioned above, normalization with soma for each cell would be sufficient.

Minor comments

5- I was intrigued by how authors obtained axonal fraction of neuronal cultures are prepared using 3 um pore sized meshes. Would the dendrites not grow towards the bottom of the mesh? Dendrites can be thicker than 3 um but some can be thinner.

6- Authors measure axon length in DIV14, I am having a hard time understanding how this was achieved. Normally, in primary neuronal cultures axons grow sampling large sections of the coverslip. I understand that mosaic images were generated, however if there are a few neurons in the same coverslip, it would be highly difficult to determine which axon belongs to which cell. How many neurons we transfected in this analysis per coverslip? Could authors explain the procedure in more detail and perhaps provide an example image of the axons that are imaged and an example of the axon trace.

7- There are two consecutive sentences starting with "while" in the introduction that authors may want to change.

Dear Editor,

Below you find our response to the remaining issues raised by the referees.

Reviewer #1:

The authors have responded satisfactorily to my suggestions. I also accept the refusal to follow one of my recommendations. For my part this work can be published in Nature Communications.

Response: We thank the reviewer for this positive evaluation of our revisions.

Reviewer #2:

The authors have performed a number of new experiments and have improved the manuscript. This is a comprehensive analyses of the relationship between a microtubule regulating enzyme and dendrite development.

I unfortunately still have reservations of this manuscript for the following reasons:

1. The molecular links between NEK7, Eg5 and MT are known. This is not a real problem for this paper since the key questions is whether this pathway is involved in dendrite branching and how it does it.

Response: We agree. In fact one of the messages of our study is that one can use information obtained in analyses of mitotic spindle assembly to learn about microtubule organization in non-mitotic cells such as differentiated neurons. However, here the context is very different and while the molecular links between NEK7 and Eg5 are maintained, their role in microtubule organization and the cellular effects are very different. This is described in our manuscript.

2. The phenotypes. In vitro, the phenotypes are consistent with each other. However, the shape of these cultured neurons are dramatically different between Fig. 1 and Fig. 8. This does raise the question about how physiological this prep is in terms of assessing dendrite development. The fact that the in vivo development phenotype is extremely subtle Fig. 8i is alarming. I requested to do more in vivo analyses but the authors argued that they couldn't get enough animals.

Response: Indeed the in vivo phenotype is less severe than in vitro. This could be for various reasons, e.g. due to technical issues, since the in vivo analysis of dendrite morphology is more difficult than in vitro, or compensatory mechanisms that may occur in animals permanently lacking NEK7 but not upon acute depletion of NEK7 by RNAi. Whatever the reason, the important point is that the dendrite defects observed in vivo are consistent with our in vitro findings. Regarding the differences in the shape of the dendritic trees in vitro vs. in vivo we would like to point out that the morphology of hippocampal neurons in vivo is typically not achieved when these cells are cultured in vitro, in the absence of the tissue context. However, such cultures are

commonly used for the analysis of general parameters of dendrite arborization including total length and branching.

3. the mechanisms. If we believe that there is a real phenotype here, then we can try to assess if the mechanism is novel and interesting. There are EB1 phenotypes in distal dendrite, but not in proximal dendrites. This difference is not satisfyingly explained. It is also not clear why retrograde MT polymerization should be inhibitory to dendrite growth, as cited in the abstract. So at the end, I felt these data still did not lead to a clear model of MT and dendrite growth.

Response: Indeed, the strongest MT polarity defects are observed in distal dendrites. Our interpretation, which we state and support with data in our manuscript (Fig. 4 and discussion, p. 22), is that Eg5 is targeted by NEK7 phosphorylation to the distal parts of dendrites, explaining why it predominantly affects MT polarity in this region. Since dendrite growth requires MT-dependent transport and since transport depends on correct MT polarity, one would expect that impairing MT stability and correct polarity in particular in the distal regions through inhibition of NEK7/Eg5 would interfere with dendrite growth. Supporting this interpretation, we also show in an independent approach that altering MT polarity by inducing ectopic nucleation through expression of the CDK5RAP2 CM1 fragment also interferes with dendrite growth and branching (Fig. 5g-l).

Reviewer #3:

This is a thorough study on the roles of Nek7 and Eg5 on dendrite and spine development, where effects of both of these proteins are characterized in cultures and Nek7's role is also studied in a knockout mouse model. Authors carefully selected Nek7 to study based on its developmentally regulated expression level. They find that Nek7 and Eg5 are required for appropriate dendrite and spine formation. The manuscript is interesting and relevant for the neuroscience audience. The translation of microtubule regulatory machinery which is initially discovered in dividing cells to postmitotic neurons is very nice. Authors provide evidence showing that Nek7 alters dendritic distribution of Eg5. Nek7's phosphorylation of Eg5 is key for dendritic distribution. Both Nek7 and Eg5 alter microtubule dynamics in distal dendrites and Eg5 activity regulates microtubule stability. The effect of STLC on Eg5's microtubule binding is supported by EOS-Eg5 phosphoconversion experiments.

Overall culture data provides evidence on Nek7 regulating microtubule dynamics and dendrite development as well as Eg5 localization and separately it provides data on Eg5's role on microtubule dynamics and microtubule stability using drugs targeting Eg5. There might be multiple effectors of Nek7 and it is not surprising that Nek7 shRNA effects are not rescued by opposing effects of FCPT. Together with S1033D mutant's rescue of dendrite deficits caused by Nek shRNA, study supports that Nek's effects are partially mediated by Eg5.

Response: We thank the reviewer for the careful evaluation of our work and the overall positive comments.

There are sections of the study that I think should be better supported with experiments and/or analysis.

Major comments

1- Authors use STLC and FCPT to alter Eg5 function. How specific are these molecules? Could they target other kinesins? They have generated Eg5 shRNA, which reduces endogenous Eg5 levels (a key reagent in the study). Does the Eg5 shRNA reduce dendrite length similar to STLC? Authors should either test the effect of Eg5 shRNA on dendrite morphogenesis (the most prominent phenotype) or authors should confirm that the effect of pharmacological treatments (STLC and FCPT) are absent in Eg5 shRNA expressing neurons, confirming their specificity.

Response: Indeed we have observed that depletion of Eg5 by shRNA also reduces dendrite length similar to STLC treatment. We have added this data to Suppl. Fig. 3 i, j.

2- Authors' hypothesis of Nek7 regulating Eg5 on Ser1033 is based on previous studies, which nicely show the phosphorylation being affected by Nek7. The Nek7 KO mice is a very useful reagent to test this finding in vivo. Authors should request the Ser1033 phosphospecific antibody and test if in mouse brain lysates the phosphorylation levels are changed or not. Phosphospecific antibodies often are not specific in immunostaining so I would recommend Westerns for this experiment.

Response: We initially tested the phosphospecific antibody mentioned by the reviewer by western blotting of human and mouse interphase and mitotic cell extracts. While we could confirm specific recognition of the phosphoepitope in human Eg5 (enriched in the mitotic sample), the antibody did not recognize the phosphoepitope of the mouse protein, presumably due to species differences in the amino acids surrounding the phosphorylation site.

3- Fig 4. The phosphomimetic mutants D or E are usually ineffective as mimics, but as in this example, rarely they act as true phosphomimics. In Figure 4a and b, authors normalized the endogenous Eg5 intensity by dividing dendrite with soma intensity. In figure 4e-f authors normalized overexpressed Eg5 intensity with the GFP intensity. I think for all quantifications the Eg5 measurements should be normalized with GFP (to control for volume changes) and also with the soma intensity (to control for the total levels of Eg5 in that particular cell). This way we can have a measure of the localization of Eg5 to dendrites. For Figure 4a-b authors did not normalize for GFP, however later on in 4d they state that there is no overall difference between different conditions in GFP dendrite/soma ratio. Therefore, it is unlikely that observed changes in 4b would be due to volume change. So, 4a-b is convincing, as it is. For 4e-f authors state that they have normalized with GFP in order to compensate for the cell to cell variability of overexpressed Eg5. However, expression levels of GFP and Eg5 are independent, a cell may have high levels of GFP and low levels of Eg5. Authors should also quantify in 4f dendrite/ soma of GFP normalized overexpressed Eg5 intensities.

Response: We have followed the reviewer's suggestion and re-analyzed the data in Fig. 4e-f by normalizing to the soma intensities. This resulted in small changes in the relative intensities between samples, but did not affect the overall result and conclusions. Fig. 4e-f now contains the new quantifications and we have also changed the legend accordingly.

4- In Fig 3 and 4, about overexpression of wt and mutant Eg5: 3a shows similar expression levels of mutants and wt and that these levels are similar to endogenous Eg5. This piece of evidence does not mean that when rescue experiments are performed in 3b-d or 4 e-f, the lipofectamine transfected neurons' Eg5 levels were similar. There is large variation in expression levels of lipofectamine transfected neurons, more variability than lentivirus expression. For Fig 3d, authors should clarify how they have selected neurons to image (did they check FLAG expression levels?) For 4e-f, as mentioned above, normalization with soma for each cell would be sufficient.

Response: We thank the reviewer for pointing this out. Fig. 3a only shows that the mutations do not affect the expression levels. Regarding variable expression levels in transfected cells: yes, we always stained neurons also with anti-FLAG antibody. This allowed us to quantify only neurons with similar expression levels and discard those with very low or very high expression, based on anti-FLAG staining intensity. We have included information about this procedure now in the legend of Fig. 3.

Minor comments

5- I was intrigued by how authors obtained axonal fraction of neuronal cultures are prepared using 3 um pore sized meshes. Would the dendrites not grow towards the bottom of the mesh? Dendrites can be thicker than 3 um but some can be thinner.

Response: To obtain dendrite and axon fractions from cultured neurons we followed the procedure described by Stuessi et al. (Nature, 2010, 327, 704-707), which also employed a 3 μm pore size. We agree, most likely some thinner dendrites also reach the bottom of the filter, so the fractionation may not be absolutely clean. However, despite this technical limitation we still see a clear enrichment of NEK7 in the somato-dendritic fraction. In contrast, Tau, an axonal marker, is present mainly in the bottom (axonal) fraction, and α -tubulin is distributed, as expected, evenly between both fractions.

6- Authors measure axon length in DIV14, I am having a hard time understanding how this was achieved. Normally, in primary neuronal cultures axons grow sampling large sections of the coverslip. I understand that mosaic images were generated, however if there are a few neurons in the same coverslip, it would be highly difficult to determine which axon belongs to which cell. How many neurons we transfected in this analysis per coverslip? Could authors explain the procedure in more detail and perhaps provide an example image of the axons that are imaged and an example of the axon trace.

Response: We have added an explanation of the procedure in the method section (p. 24). We used low-density cultures for this analysis and typically obtained 10-15

transfected neurons per coverslip. Below is an example image of an individual neuron with traced axon.

14DIV Neuron axon (GFP)

Axon tracing

Merge

7- There are two consecutive sentences starting with “while” in the introduction that authors may want to change.

Response: We thank the reviewer for noticing this. We have changed the text.